# Witnessing light-driven entanglement using time-resolved resonant inelastic X-ray scattering

Jordyn Hales[1,5], Utkarsh Bajpai[1,5], Tongtong Liu[2], Denitsa R. Baykusheva [3], Mingda Li [4], Matteo Mitrano[3] ✉ & Yao Wang [1] ✉

Characterizing and controlling entanglement in quantum materials is crucial for the development of next-generation quantum technologies. However, defining a quantifiable figure of merit for entanglement in macroscopic solids is theoretically and experimentally challenging. At equilibrium the presence of entanglement can be diagnosed by extracting entanglement witnesses from spectroscopic observables and a nonequilibrium extension of this method could lead to the discovery of novel dynamical phenomena. Here, we propose a systematic approach to quantify the time-dependent quantum Fisher information and entanglement depth of transient states of quantum materials with time-resolved resonant inelastic x-ray scattering. Using a quarter-filled extended Hubbard model as an example, we benchmark the efficiency of this approach and predict a light-enhanced many-body entanglement due to the proximity to a phase boundary. Our work sets the stage for experimentally witnessing and controlling entanglement in light-driven quantum materials via ultrafast spectroscopic measurements.

Quantum materials are systems featuring collective electronic behavior[1] with broad technological applications, such as superconductivity[2], topological order[3], and quantum spin liquidity[4,5]. Underlying these emergent phenomena is the presence of entanglement among subparts of the electronic wavefunctions[6,7], which has important fundamental and applied consequences. On one hand, quantum fluctuations caused by entanglement play an important role in the appearance of quantum phase transitions characterized by unconventional behavior[8–13]. On the other hand, entanglement constitutes a precious resource for material-based quantum computing, where information is encoded and manipulated via arbitrary entangled or even multipartite entangled states[14,15]. Therefore, accurately characterizing and controlling entanglement in the solid state is a key step towards the realization of future quantum technologies[16].

The many-body wavefunction of a highly entangled quantum system cannot be expressed as the direct product of multiple single-particle states in any basis. Entangled wavefunctions in synthetic few-body quantum simulators can be experimentally characterized[15,17–20] through the Rényi entropy[21–25] and multi-point correlations[26–33]. However, the measurement complexity increases with the Hilbert-space dimension and scales exponentially with the system size. As solid-state measurements are restricted to a limited number of macroscopic observables, a tomography of electronic wavefunctions in quantum materials becomes impractical[34–37].

A more efficient approach for investigating entanglement in real materials relies on determining the "entanglement depth"[38–41], defined as the minimum number of entangled modes required to construct a specific many-body state. The bounds on entanglement depth of a quantum system can be quantified through expectation values of

[1]Department of Physics and Astronomy, Clemson University, Clemson, SC 29634, USA. [2]Department of Physics, Massachusetts Institute of Technology, Cambridge, MA 02139, USA. [3]Department of Physics, Harvard University, Cambridge, MA 02138, USA. [4]Department of Nuclear Science and Engineering, Massachusetts Institute of Technology, Cambridge, MA 02139, USA. [5]These authors contributed equally: Jordyn Hales, Utkarsh Bajpai. ✉ e-mail: mmitrano@g.harvard.edu; yaowang@g.clemson.edu

specific operators called entanglement witnesses[42-46]. Recent developments along this direction have been enabled by use of the quantum Fisher information (QFI)[47-49], which can be quantified from equilibrium spectroscopy[50]. By extracting the QFI from the imaginary part of the dynamical susceptibility, it is possible to witness a lower bound for the entanglement depth without relying on a full tomography of the wavefunction[51-53]. This approach has been successfully applied to the study of magnetic materials[54-56], demonstrating its feasibility in solid-state experiments at equilibrium.

Diagnosing entanglement in quantum materials out of equilibrium would be particularly impactful, as ultrafast lasers have led to the synthesis of nontrivial states of matter without equilibrium analogues[57-59]. Dynamical entanglement has been theoretically demonstrated in the wavefunctions of solvable toy models[44,60,61], but so far it has been unclear how to experimentally characterize this phenomenon in nonequilibrium spectroscopic experiments. Inspired by quantum metrology, entanglement can be quantified by the fluctuations of local probes at a given distance or a finite momentum, and the recently developed time-resolved resonant inelastic x-ray scattering (trRIXS) technique[62-64]—sensitive to collective charge, orbital, spin, and lattice excitations[62,65-68]—opens new opportunities for diagnosing entanglement in light-driven systems.

Witnessing entanglement with trRIXS requires measuring the scattering cross-section of a light-driven material, evaluating the transient dynamical structure factors, calculating the QFI, and determining relevant quantum bounds signaling the presence of entanglement for a certain observable (see Fig. 1). Each step encapsulates a distinct challenge, and we illustrate them by focusing on the spin degrees of freedom. First, estimating accurate spin fluctuations using trRIXS requires separating the contributions of high-order excitations mediated by the intermediate state of the scattering process. Second, competing time and energy resolution effects imply that the QFI for the instantaneous wavefunctions[69] cannot be directly extracted from the spectrum at a single time delay and must be deconvolved on the time axis. Third, assessing the entanglement depth requires calculating

suitable theoretical bounds for the QFI operator. While the connection between QFI and multiparticle entanglement is well established for spin operators in magnetic materials and gapped fermionic lattice modes[70], this connection is unclear for interacting fermions with dopant carriers, which is particularly relevant to the study of light-driven materials.

Here, we address these challenges and develop a procedure to witness entanglement using trRIXS. As a brief summary of the main results, we quantify the impact of the finite core-hole lifetime on the evaluation of dynamical structure factor, develop a self-consistent protocol to calculate the instantaneous QFI, and assess the multipartite entanglement dynamics using the QFI and appropriate theoretical bounds. Through this procedure, we focus on the study of light-induced entanglement when a system is close to a phase boundary and choose a one-dimensional (1D) extended Hubbard model (EHM). The EHM under consideration, with repulsive on-site interaction and attractive nearest-neighbor interaction, has been recently identified as the underlying microscopic description of 1D cuprate chains (e.g., $Ba_{2-x}Sr_xCuO_{3+\delta}$)[71]. By comparing its dynamics with that of a pure Hubbard model, we find that the nonlocal interaction plays a crucial role on the nonequilibrium enhancement of many-body entanglement.

## Results
### Quantifying the time-dependent quantum Fisher information from trRIXS

For a time-dependent wavefunction $|\psi(t)\rangle$ and in the Schrödinger picture, the instantaneous QFI density associated with a local spin operator is defined as[47-49]

$$f_Q(q,t) = \frac{4}{N} \sum_{ij} e^{iq(r_i-r_j)} \left[ \langle\psi(t)|S_i^z S_j^z|\psi(t)\rangle \right.$$
$$\left. - \langle\psi(t)|S_i^z|\psi(t)\rangle \langle\psi(t)|S_j^z|\psi(t)\rangle \right]. \quad (1)$$

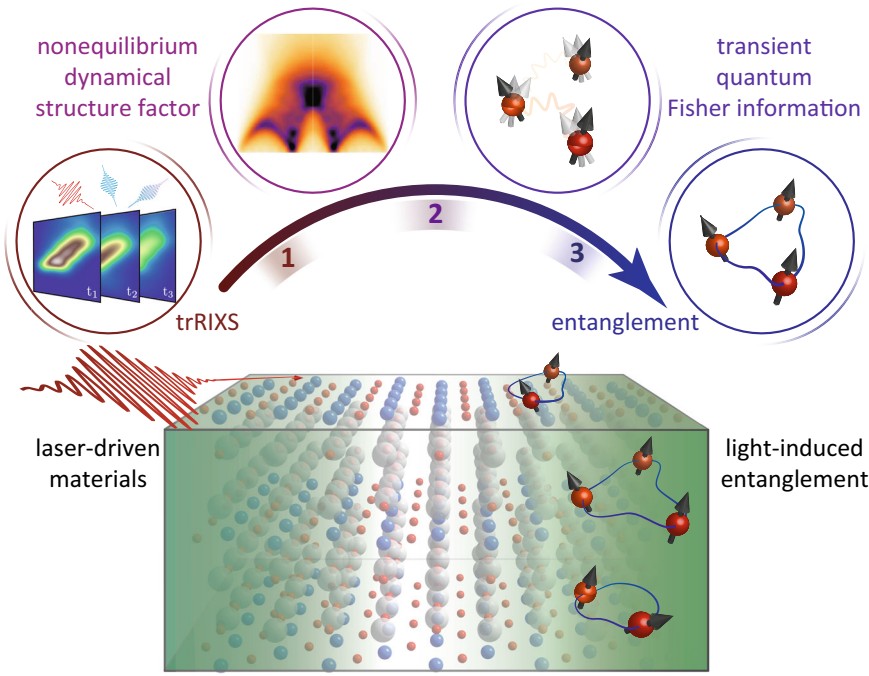

**Fig. 1 | Probing light-driven entanglement in quantum materials.** An intense pump laser drives a material out of equilibrium and its time-dependent collective excitations are probed by time-resolved resonant inelastic X-ray scattering (trRIXS). One witnesses many-body entanglement by first extracting the nonequilibrium dynamical structure factor from the trRIXS response function, then calculating the quantum Fisher information associated with a specific operator using the transient dynamical structure factor, and, finally, comparing the transient quantum Fisher information with operator-specific quantum bounds.

where $r_i$ is the real-space position vector of site $i$, $S_j^z$ is the localized spin operator ($z$ component), and $N$ is the total number of sites. The transient QFI $f_Q(q, t)$ is completely determined by the instantaneous wavefunction $|\psi(t)\rangle$, or an ensemble represented by the density matrix $\rho(t)$. However, different from the integral relation for equilibrium states[50], one cannot obtain the $f_Q(q, t)$ by directly integrating an instantaneous pump-probe spectrum at a certain time delay $t$, due to the convolution with finite-width probe pulses. Such a finite width contributes to the energy resolution of time-resolved spectroscopy experiments and is usually comparable to the natural timescales of the investigated systems. We describe here a strategy to extract the time-dependent QFI from a sequence of nonequilibrium spectra.

Let us first assume that the dynamical structure factor $S(q, \omega, t)$ is experimentally accessible, leaving the discussion of its extraction from x-ray measurements to section "Time-resolved resonant inelastic X-ray scattering." The $S(q, \omega, t)$ is defined as[72]

$$S(q, \omega, t) = \frac{1}{2\pi N} \sum_{ij} e^{-iq(r_i - r_j)} \iint_{-\infty}^{+\infty} dt_1 dt_2\, g(t_1; t) g(t_2; t) \left\langle \hat{S}_i^z(t_1) \hat{S}_j^z(t_2) \right\rangle e^{i\omega(t_1 - t_2)}, \quad (2)$$

where $g(\tau; t)$ denotes the temporal probe envelope, usually approximated with a Gaussian profile[73]

$$g(\tau; t) = \frac{1}{\sigma_{pr}\sqrt{2\pi}} e^{-(\tau - t)^2 / 2\sigma_{pr}^2}, \quad (3)$$

with pulse duration $\sigma_{pr}$. This duration is physical, primarily set by the spectral content of the laser pulses but also renormalized by the instruments and the material's self-energy. We express the pump-probe spectra in the interaction picture (same below), where the operator $\hat{\mathcal{O}}(t) = \mathcal{U}(-\infty, t)\mathcal{O}\,\mathcal{U}(t, -\infty)$ evolves via the unitary operator $\mathcal{U}(t, t_0) = \hat{\mathcal{T}}_t[e^{-i\int_{t_0}^{t} \mathcal{H}(t')dt'}]$. Here, the time-dependent Hamiltonian $\mathcal{H}(t)$ only includes the pump (and not the probe) field.

Due to the effects of the probe pulse profile on time and energy resolution, the spectrum $S(q, \omega, t)$ at time $t$ is determined not only by the instantaneous wavefunction $|\psi(t)\rangle$, but also by earlier or later wavefunctions in a finite time window. Therefore, the QFI density $f_Q(q, t)$, which diagnoses the entanglement of the instantaneous wavefunction $|\psi(t)\rangle$, cannot be evaluated by simply integrating the $S(q, \omega, t)$ along the energy axis. As we show in Supplementary Note 1, the relationship between the time-dependent QFI $f_Q(q, t)$ and the transient structure factor $S(q, \omega, t)$ becomes an implicit integral equation

$$\int_{-\infty}^{+\infty} d\tau\, g(\tau; t)^2 f_Q(q, \tau) = 4 \int_{-\infty}^{+\infty} d\omega\, S(q, \omega, t). \quad (4)$$

Note that we have assumed the absence of long-range magnetic order at the specific momentum $q$, which is the case for the simulations in this paper. If a long-range order is present, one should further subtract the elastic peak from the structure factor, whose intensity corresponds to the disconnected part (second term) of Eq. (1). In the limit of ultrashort probe pulses, i.e., $\sigma_{pr}$ smaller than any nonequilibrium physical timescale of the system, the envelope $g(\tau; t)^2$ can be approximated by a delta function $\delta(\tau - t)$, leading to an explicit solution consistent with the equilibrium sum-rule integral[50]. However, when the probe pulse has a finite time duration (as in spectroscopy experiments with high energy resolution), this approximation breaks down. In order to extract the instantaneous QFI from this implicit Eq. (4), we expand its left-hand side and convert the equation into a self-consistent integro-

differential problem. As detailed in Supplementary Note 1, this leads to

$$f_Q(q, t) = 8\sigma_{pr}\sqrt{\pi} \int_{-\infty}^{+\infty} d\omega\, S(q, \omega, t) + \sum_{m=1}^{\infty} \frac{\mathcal{C}_m}{(2m)!} \frac{\partial^{2m} f_Q}{\partial t^{2m}}, \quad (5)$$

where $\mathcal{C}_m = -(1/\sigma_{pr}\sqrt{\pi}) \int_{-\infty}^{\infty} e^{-x^2/\sigma_{pr}^2} x^{2m} dx = -(\sigma_{pr}^{2m}/\sqrt{\pi})\Gamma(m + 1/2)$. In the presence of time-translational invariance, the infinite series on the right-hand side of Eq. (5) vanishes and we reproduce the equilibrium relation $f_Q^{eq}(q, \tau) = 8\sigma_{pr}\sqrt{\pi} \int_{-\infty}^{+\infty} d\omega\, S^{eq}(q, \omega, t)$[50]. However, high-order derivative terms on the right-hand side can play a significant role far from equilibrium, as discussed in section "TrRIXS and QFI in a driven extended Hubbard model."

The self-consistently calculated $f_Q(q, t)$ serves the purpose of witnessing entanglement in a transient $k$-partite quantum state when exceeding its operator-specific boundary[50]. While we use the pure-state notation in the derivation of the QFI sum rule and choose a pure initial state in our simulations, this approach applies to both pure and mixed initial states (see Supplementary Note 1 for further details). This generalization relies on considering $\langle \cdots \rangle$ as an ensemble average and on the linearity of Eqs. (4) and (5). We note that, as specified by Hauke et al. in ref. 50, this protocol requires a careful normalization of the scattering cross-section into absolute values by properly accounting for the trRIXS matrix elements.

## Time-resolved resonant inelastic X-ray scattering

Recent experimental work on diagnosing entanglement in the solid state focused on inelastic neutron scattering of low-dimensional spin systems[54–56], as the neutron scattering cross section is directly proportional to the dynamical spin structure factor[74]. While there is not yet an ultrafast incarnation of inelastic neutron scattering, the recent development of trRIXS provides an alternative pathway to access nonequilibrium dynamical structure factors of spin and charge degrees of freedom[62–65]. As detailed in Methods, trRIXS is a photon-in-photon-out x-ray scattering process involving an intermediate state with a finite lifetime. Due to the spin-orbit coupling at the core level (e.g., the $2p$ orbitals for the transition-metal $L$-edge RIXS), this intermediate state can involve spin flip events and couple to magnetic excitations of the valence band[75]. Therefore, trRIXS is sensitive to spin excitations and can be used to probe the nonequilibrium spin dynamics of light-driven materials[76].

The trRIXS cross-section, denoted as $\mathcal{I}(q, \omega_i - \omega, \omega_i, t)$ in Methods, depends on energy, momentum, and polarization of both the incident and scattered photons. In our simulation, we select a scattering geometry with $\pi$-polarized (parallel to the scattering plane) incident photons and $\sigma$-polarized (perpendicular to the scattering plane) scattered photons, which maximizes the spin-flip contribution to the trRIXS cross-section[75,76]. Due to our focus on spin entanglement, we keep this polarization configuration fixed throughout the paper and omit the polarization subscripts in $\mathcal{I}$. The trRIXS spectrum comes with two energy axes for the incident photon energy $\omega_i$ and the energy loss $\omega$ (difference between incident and scattering photon energies). Figure 2 shows sample trRIXS spectra for a driven extended Hubbard model (see section "TrRIXS and QFI in a driven extended Hubbard model") and for a range of incident energies $\omega_i$. We select the resonance $\omega_i$ by maximizing the trRIXS intensity. At fixed $\omega_i$, the nonequilibrium dynamical structure factor $S(q, \omega, t)$ can be estimated by

$$S(q, \omega, t) \approx \frac{\mathcal{I}(q, \omega_i - \omega, \omega_i, t)}{\tau_{core}^2 |M_{q_i\varepsilon_i}^{(in)} M_{q_s\varepsilon_s}^{(out)}|^2}, \quad (6)$$

which implies replacing the excitation operator $\hat{\mathcal{D}}_{q_s\varepsilon_s}^\dagger(t_1')\hat{\mathcal{D}}_{q_i\varepsilon_i}(t_1)$ with a spin-flip operator $\hat{S}^+(t_1)$. Here, $\hat{\mathcal{D}}_{q_i\varepsilon_i}$ and $\hat{\mathcal{D}}_{q_s\varepsilon_s}^\dagger$ denote the dipole-transition operators for the incident and scattering processes, respectively; $M_{q_i\varepsilon_i}^{(in)}$ and $M_{q_s\varepsilon_s}^{(out)}$ indicate the corresponding matrix

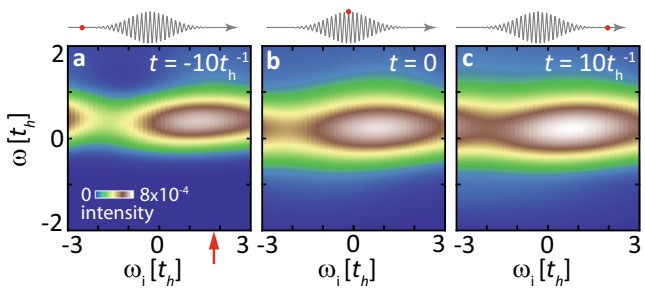

**Fig. 2 | Overview of time-resolved Resonant Inelastic X-ray Scattering (trRIXS) spectra. a**–**c** Snapshots of the trRIXS spectra with $q = \pi/6$ and for selected pump-probe time delays. The red arrow indicates the resonance incident energy $\omega_i$ used to analyze the structure factors. The upper insets show the time relative to the pump pulse, whose amplitude is $A_0 = 1$ and frequency is $\Omega = 10t_h$. The core-hole lifetime is set as $\tau_{core} = 1/3\,t_h$.

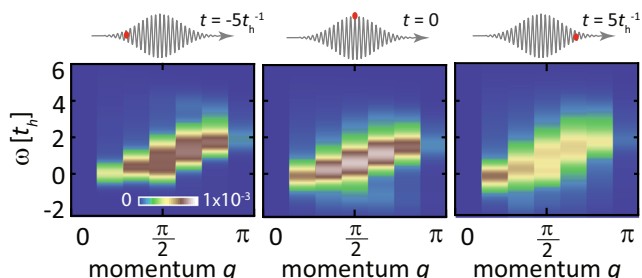

**Fig. 3 | Momentum distribution for time-resolved Resonant Inelastic X-ray Scattering (trRIXS) spectra.** From left to right: trRIXS spectra for the 1D momentum $q$ ranging from 0 to $\pi$, at $t = -5t_h^{-1}$, $t = 0$, and $t = 5t_h^{-1}$, respectively. The incident energy $\omega_i$ is set as $1.8t_h$ and the upper insets show the time relative to the pump pulse, whose condition is the same as Fig. 2.

elements and $\tau_{core}$ is the core-hole lifetime. The definition of these factors and the full expression of the trRIXS cross-section $\mathcal{I}(q, \omega_i - \omega, \omega_i, t)$ are explained in the Methods section. The estimation of Eq. (6) is analogous to the ultrashort core-hole lifetime (UCL) approximation in equilibrium[75,77], and it can be proven that Eq. (6) becomes exact in the $\tau_{core} \to 0$ limit.

### TrRIXS and QFI in a driven extended Hubbard model

In this work, we aim to witness entanglement with trRIXS in a proto-typical correlated electron system. Given prior equilibrium RIXS experiments[78], we consider 1D cuprate chains as an ideal platform for the experimental verification of our results. Recent experiments in $Ba_{2-x}Sr_xCuO_{3+\delta}$ have identified the EHM with mixed-sign interactions[71] as their underlying model Hamiltonian (see Methods) and here we investigate its light-driven dynamics. Throughout this paper, we set the on-site ($U$) and the nearest-neighbor ($V$) interactions to $U = 8t_h$ and $V = -t_h$, respectively, corresponding to the characteristic values for cuprate chains[71,79]. The existence of this nearest-neighbor term $V$ is crucial for the presence of many-body entanglement, as we discuss in section "Light-enhanced entanglement."

We introduce the pump excitation in our second-quantized electrons through the standard Peierls substitution. The pump laser pulses are described by a vector potential in the form of an oscillatory Gaussian $A(t) = A_0\, e^{-t^2/2\sigma_{pump}^2} \cos(\Omega t)$ with fixed width $\sigma_{pump} = 3t_h^{-1}$, variable amplitude $A_0$, and frequency $\Omega$. The ground state of the EHM is calculated by the parallel Arnoldi method with Paradeisos acceleration[80,81] and the time evolution is evaluated by the Krylov subspace technique[82,83]. We adopt a 12-site chain with periodic boundary conditions and quarter filling throughout this paper, due to its proximity to the triplet-pairing phase[84,85]. We employ the ground state at zero temperature as the initial state, due to the computational complexity of simulating the trRIXS cross-section of an ensemble. The generality of this method is further discussed in "Quantifying the time-dependent quantum Fisher information from trRIXS" and the SI.

Figure 2a shows selected theoretical $\pi$-$\sigma$-polarized trRIXS spectra. The incident energy $\omega_i = 1.8t_h$ (defined relative to the absorption edge) is determined according to the equilibrium resonance profile for the singly-occupied initial state. By varying the momentum transfer $q$, the equilibrium RIXS spectrum of a quarter-filled EHM displays a two-spinon continuum ($t = -5t_h^{-1} < -\sigma_{pump}$), as shown in Fig. 3. Different from undoped antiferromagnets[56], low-energy spin excitations of a doped extended Hubbard model mainly lie near the nesting vector $q = 2k_F$ of spinon Fermi surface[86]. At large momenta ($q \geq \pi/2$), the trRIXS spectral weights are gradually suppressed by the X-ray scattering matrix elements[77,87], which we divide out as shown in Eq. (6) when evaluating the spin QFIs. When calculating the time-

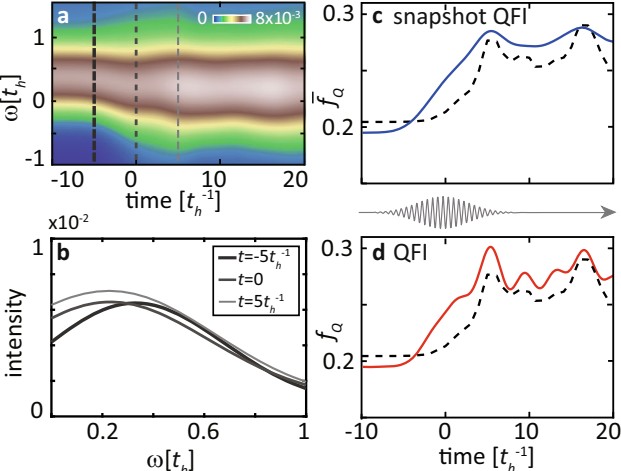

**Fig. 4 | Nonequilibrium dynamical structure factor and transient quantum Fisher information (QFI). a** Evolution of the dynamical spin structure factor $S(q, \omega, t)$ estimated from $\mathcal{I}(q, \omega_i - \omega, \omega_i, t)$ by fixing the incident energy at $\omega_i = 1.8t_h$, as in Fig. 2. **b** Spectral distribution for $t = -5t_h^{-1}$, 0, and $5t_h^{-1}$, respectively. **c** Time-dependence of the snapshot quantum Fisher information (QFI) density $\bar{f}_Q(q,t)$ (blue curve) evaluated by integrating the instantaneous spectral cuts [e.g., each curve in **b**] and of the exact QFI (black dashed line). **d** Time dependence of the QFI density $f_Q(q, t)$ (red curve) evaluated by the self-consistent iteration in Eq. (5) and comparison with the exact QFI (black dashed line). The core-hole lifetime for these data is $\tau_{core} = 1/3\,t_h$, while the momentum transfer is fixed to $q = \pi/6$.

dependent QFI, we focus on a specific momentum, $q = \pi/6$. Distinct from undoped antiferromagnets previously employed to witness equilibrium entanglement[56], this small-momentum wavevector exhibits the most evident spectral changes and captures the longest-range correlations supported by our system size. The 1D system simulated in this paper does not have a spontaneous symmetry breaking, which simplifies the entanglement analysis of the spin fluctuations.

At the selected resonance and momentum transfer, we calculate the time-dependent $S(q, \omega, t)$ from the trRIXS intensity following Eq. (6), as shown in Fig. 4a. At the center of pump pulse, i.e., $t = 0$, the excitation spectrum experiences an overall softening [also see Fig. 4b] due to a Floquet renormalization of the spin-exchange energy, and a slight broadening of the spectral peak[76,88]. These spectral changes persist long after the pump pulse as a result of strong correlation effects[72]. However, different from the case of light-driven spin spectra at the top of a magnon band[76], the softening here is accompanied by an

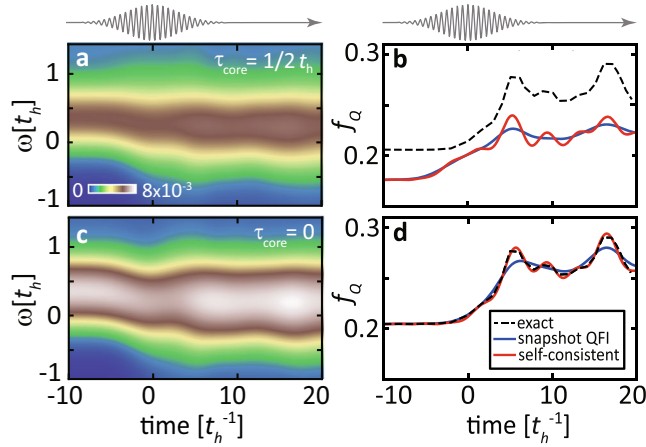

**Fig. 5 | Core-hole lifetime contribution to the quantum Fisher information (QFI) evaluation. a** Evolution of $S(q, \omega, t)$ extracted from the trRIXS intensity for core-hole lifetime set as $\tau = 1/2t_h$. **b** Comparison of the exact results of instantaneous QFI (black dashed line), the snapshot QFI density $\bar{f}_Q(q, t)$ (blue curve) evaluated the direct integration, and the QFI density $f_Q(q, t)$ (red curve) evaluated by the self-consistent iteration. **c**, **d** Same as **a**, **b** but for the zero core-hole lifetime situation, where trRIXS is identical to $S(q, \omega, t)$. The upper insets show the evolution of the pump-field vector potential.

increase of the spectral intensity. To analyse the nonequilibrium entanglement, the evolution of the intensity is more relevant than that of the peak position.

To test our framework for evaluating the transient QFI of light-driven spin degrees of freedom, we first calculate the exact values of $f_Q(q, t)$ [dashed curves in Fig. 4c, d] by plugging the simulated time-dependent wavefunctions $|\psi(t)\rangle$ into Eq. (1). Note that this direct evaluation is theoretically possible due to the access to the instantaneous wavefunctions, which cannot be measured in experiments. We then proceed to evaluate the QFI as it would be done in real trRIXS experiments, i.e., by only assessing the sequence of trRIXS snapshots without additional theoretical knowledge about the state of the driven system. A direct extension of the equilibrium formula in ref. 50 entails treating each single time delay as an equilibrium spectrum. To distinguish it from the nonequilibrium definition in Eq. (5), we rename it as "snapshot QFI"

$$\bar{f}_Q(q, t) = 8\sigma_{pr}\sqrt{\pi} \int d\omega\, S(q, \omega, t). \qquad (7)$$

Compared with the exact QFI densities, the $\bar{f}_Q(q, t)$ (blue curve) overestimates the transient increase near $t \cdot 0$ and does not capture the oscillations after the pump ends [see Fig. 4c]. Such a deviation at ultrafast timescales reflects the presence of convolution effects caused by the finite probe width.

The failure of the snapshot QFI in capturing the exact QFI evolution requires the introduction of the full self-consistent iteration Eq. (5). Compared to the snapshot QFI, the latter contains high-order time-derivatives, which are non-negligible when the spectrum varies rapidly in time. We evaluate these high-order derivatives using finite difference methods, starting from the time sequence of trRIXS spectra, and then solving the self-consistent equations. This procedure leads us to the $f_Q(q, t)$ indicated by the red curve in Fig. 4d. The full self-consistent QFI is closely aligned with the exact QFI behavior and captures the time-dependent oscillations induced by the pump for $t > 5t_h^{-1}$ [see Fig. 4d]. This implies that the self-consistent calculation of the QFI is essential for capturing fast coherent dynamics and it is not sufficient to approximate each time delay as a quasi-equilibrium spectrum.

The remaining deviation with respect to the exact evolution of QFI, which occurs both at equilibrium and at the center of the pump, can be attributed to the fact that the trRIXS spectrum is not identical to the spin structure factor $S(q, \omega, t)$. Such a discrepancy is known in equilibrium RIXS, which captures the poles of $S(q, \omega, t)$ but is less accurate in yielding its spectral weight[77]. Physically, the intermediate state in trRIXS has a finite lifetime and contains additional dynamics, such as multi-magnon or spin-charge excitations, besides instantaneous spin-flip events in the UCL limit. Due to the admixture with these excitations, Eq. (6) is only a good approximation but not an identity. The trRIXS spectrum underestimates the QFI at equilibrium, where the local moment is maximal and other processes are irrelevant (see Supplementary Fig. 2), while overestimating the QFI at the center of the pump pulse where there are more charge carriers induced by the laser.

To quantify the effect of a finite core-hole lifetime, we compare simulations with $\tau_{core} = 1/2t_h$ and $\tau_{core} = 0$. As shown in Fig. 5, the QFIs extracted from the trRIXS spectra converge towards the exact calculations with decreasing core-hole lifetime. In the limit of an infinitesimal $\tau_{core}$ [see Fig. 5d], the self-consistently calculated $f_Q(q, t)$ precisely matches the exact QFI density obtained from the instantaneous wavefunctions. In contrast, the "snapshot QFI" $\bar{f}_Q(q, t)$ (blue curve) still deviates from them, reflecting the intrinsic error caused by the finite time resolution due to the probe pulse. When the lifetime is not negligible, the conversion of the equilibrium RIXS intensity into the $S(q, \omega)$ requires a more systematic approach. One method consists of calculating the four-particle response function as the lowest-order perturbative expansion of the lifetime $\tau_{core}$[77]. For trRIXS, one can also correct for the finite lifetime effects by using an overall scaling factor determined by the equilibrium RIXS intensity and $S(q, \omega)$, both of which can be independently measured. As discussed in Sec. II of the SI, this correction provides a good approximation of the long-term dynamics even for a large core-hole lifetime.

Apart from accurately describing the wavefunction entanglement encoded in the pump-probe spectrum, the self-consistent Eq. (5) also provides an alternative way to interpret the light-induced spin fluctuation dynamics. We note that the leading order $m = 1$ term in the series of Eq. (5) is the second-order time derivative of $f_Q$, which reflects an inertia of the underlying wavefunction toward the change. This inertia enables self-driving of the spin fluctuations and explains why the QFI in Fig. 5 continues to oscillate for $t \gtrsim 5t_h^{-1}$ even when the pump field vanishes. In contrast, trRIXS spectra for non-interacting fermions (and accordingly any quantities extracted from them) completely recover to the initial equilibrium spectra after the pump is gone[89,90]. Therefore, the self-driving wavefunctions are a unique feature of correlated systems with interactions and including these high-order time-derivatives in Eq. (5) is crucial to correctly capture the nonequilibrium QFI dynamics. The "snapshot QFI" $\bar{f}_Q(q, t)$, on the other hand, serves as a good approximation for the $f_Q$ at the pump arrival, but starts to deviate at later time delays where the pump tails off.

Following the pump pulse, the trRIXS spectral weight at large momenta and high energies is transferred into small momenta [see Fig. 3]. This transfer suggests the onset of long-wavelength spin fluctuations, although the overall magnetic moment decreases with the generation of doublon-hole fluctuations. While we focus on the smallest momentum $q = \pi/6$ of the cluster in this work, the efficiency of the self-consistent approach is not restricted to any specific momentum. As shown in Supplemental Note 3, the evaluated QFIs accurately match those calculated through the wavefunction evolution. Therefore, the time-dependent QFIs for different momenta witness the transfer of entanglement at different length scales.

### Light-enhanced entanglement

A reliable extraction of time- and momentum-resolved QFI $f_Q(q, t)$ allows us to witness the entanglement depth of the driven EHM.

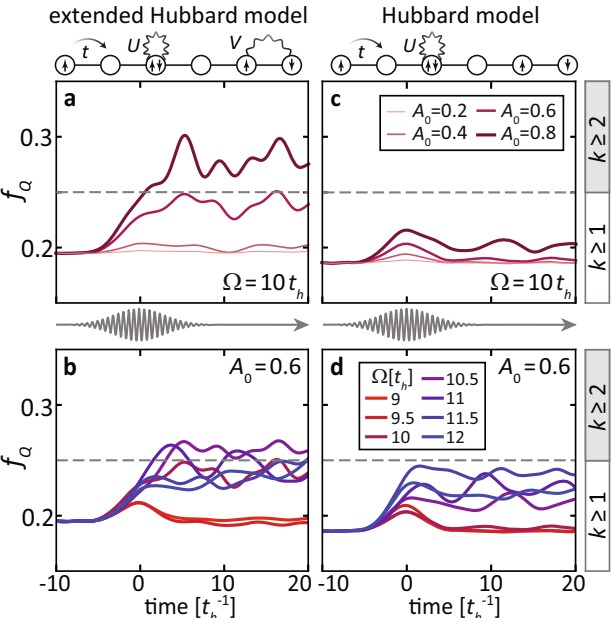

**Fig. 6 | Light-manipulation of the quantum Fisher information (QFI) for various pump conditions and models. a, b** The transient QFI density $f_Q$ in a light-driven EHM [i.e., Eq. (11) with $V = -t_h$] for selected **a** pump amplitudes at fixed energy $\Omega = 10t_h$ and **b** pump energies at fixed amplitude $A_0 = 0.6$. **c, d** Same as **a, b** but for a Hubbard model without nonlocal interactions [i.e., $V = -t_h$]. The bar to the right denotes the range and boundary for multiparticle entanglement with a $k$-producible state. The upper insets are cartoons of the Hamiltonian terms contained in each model, while the middle insets show the time evolution of the pump vector potential.

---

As originally investigated in pure spin systems[48,49,91], the QFI informs us about the presence of an entangled many-body state if its value exceeds a minimum value derived from the quantum Cramér–Rao bound. To witness entanglement dynamics in the spin sector, we determine a quantum bound suited to diagnose multipartite entanglement after obtaining the time-dependent QFI. Since the doped fermionic model has a local magnetic moment $\langle m_z^2 \rangle < 1$, we normalize the QFI bound by reducing the total spin $S$ by the doping concentration. This implies that, for a $k$-producible state, the QFI obtained by a RIXS spectrum is bounded by

$$f_Q(q, t) \leq 4kn^2 S^2 \qquad (8)$$

where $n = \langle \sum_{io} n_{io} \rangle / N$ is the average electron density per site and $S = 1/2$ in the single-band system. Here, we neglected the reduction of the normalization factor resulting from double occupation (which was discussed in refs. 92,93) since the latter is minimal in our quarter-filled system [see Supplementary Note 3]. It follows that the upper bound of $f_Q$ for a 1-producible state ($k = 1$) is 0.25, as the average density is $n = 0.5$ at quarter filling, and any value of $f_Q > 0.25$ signals the presence of at least bipartite entanglement (i.e., $k \geq 2$).

As shown in Fig. 6a, the QFI $f_Q$ increases following the excitation with a $\Omega = 10t_h$ laser pulse. As the pump amplitude increases, $f_Q$ is monotonically pushed towards the boundary and exceeds 0.25 for $A > 0.6$ (i.e., a 50% enhancement), signaling that the pump-induced state is at least bipartite entangled. Note that time evolution is non-monotonic, indicating the presence of oscillations of the many-body states which are not governed by thermalization. (This has been demonstrated by comparing nonequilibrium and finite-temperature spectra in ref. 72.)

We now investigate the pump frequency dependence of the light-enhanced entanglement. As shown in Fig. 6b, the light-induced changes are weak for $\Omega < 10t_h$. Since the particle-hole excitations in our EHM cost $U - 2V = 10t_h$, low-frequency pump pulses do not have enough energy to create doublon-hole pairs and scramble the spin configurations. Thus, the transient increase of $f_Q$, which disappears when the pump is over, can be attributed to Floquet engineering of the spin exchange interaction via virtual processes[88]. The maximal enhancement is achieved for $\Omega \sim 10 - 11t_h$, resonant with the doublon-holon excitation energy. In this case, long-wavelength spin fluctuations are generated through the creation and annihilation of doublon-hole pairs, which contribute to a more entangled many-body state. This process is slightly reduced when the pump photon energy is higher than the doublon-holon excitation energy.

We argue that the light-enhanced entanglement likely reflects the proximity to a quantum phase transition. Although our trRIXS simulation is restricted to a small cluster and cannot rigorously determine phase boundaries, we compare the QFI dynamics under the same excitation conditions in the EHM and in a doped Hubbard model (without the attractive $V$). While the two Hamiltonians only differ by a $V \sim -t_h$ term, much smaller than the dominant on-site interaction $U = 8t_h$, previous studies have shown the presence of a triplet superconducting phase for moderate attractive near-neighbor interactions[84,94–97]. This phase transition boundary has been explored through exact numerical methods in the thermodynamic limit, suggesting that, for $U = 8t_h$, triplet superconductivity develops for $-1.7t_h \lesssim V \lesssim -1.1t_h$[85]. Therefore, the EHM with $V = -t_h$ is expected to display a stronger light-induced entanglement, compared with a pure Hubbard model (without $V$), due to the proximity to this phase boundary. As shown in Fig. 6c, d, the QFI of the driven Hubbard model is much smaller than that of the EHM at same pump conditions, consistent with the absence of such quantum phase transition. In other words, the nearest-neighbor interaction, whose feature was recently identified in 1D cuprate chains but the phonon-mediated mechanism widely exists in transition-metal oxides, is crucial towards achieving light control of entanglement in correlated materials.

While here we mainly aim to witness spin-mode multipartite entanglement, one might also explore entanglement depth and quantum bounds associated with fermions. In Supplementary Note 5, we report an extension of our calculations to the single-particle fermionic modes of the driven EHM following the approach in ref. 70, with the $U(1) \times U(1)$ symmetry. The obtained bounds are weaker than Eq. (8) and they could be constrained by additional fermionic symmetries and more complex fermionic operators [see discussion in Supplementary Note 5]. A comprehensive study of basis-independent entanglement witnesses for indistinguishable fermions is beyond the scope of this work.

## Discussion

In this work, we have connected the trRIXS spectrum of a light-driven material to the entanglement depth of its time-dependent wavefunction. Our calculations explicitly account for the experimental time resolution and finite core-hole lifetime. We have developed a self-consistent procedure to extract the time-dependent QFI from a realistic pump-probe spectrum. With the full information about the x-ray probe pulse, one can reproduce the ultrafast dynamics of equal-time observables beyond the limitations set by the time resolution. The core-hole lifetime is an intrinsic property of materials and introduces a quantitative but not qualitative deviation between the extracted QFI and the exact values, which can be corrected by using equilibrium spectroscopy data.

Through the time-dependent QFI, we determine a small-wavevector enhancement of multipartite entanglement in the driven state of the EHM, which becomes at least bipartite entangled after the pump arrival. This finding is in contrast with the time evolution of a

simple Hubbard model with the same pump parameters, in which the QFI never exceeds the minimum bound for a separable state. We interpret this difference in terms of light-enhanced quantum fluctuations due to the EHM proximity with a phase transition boundary between Luttinger liquid and triplet superconductivity. The predicted enhanced entanglement depth could be measured via future trRIXS experiments on doped quasi-1D cuprates (e.g., $Ba_{2-x}Sr_xCuO_3$) and other low-dimensional correlated oxides. Due to the crucial role of the attractive interactions on the light-enhanced entanglement, these experiments will in turn help detecting nonlocal interactions, usually challenging to characterize in quantum materials. Additionally, since the nonequilibrium control of entanglement was recently discussed also in the context of spintronic devices[98], our self-consistent trRIXS approach could eventually be applied to probe time-dependent entanglement in non-optically-driven systems.

Beyond experimentally diagnosing transient entanglement dynamics in driven quantum materials, we anticipate the need for further theoretical developments. Here, we mainly discuss multipartite entanglement and quantum bounds in the spin basis. However, it is also important to characterize the entanglement depth of fermionic degrees of freedom via observables specific to indistinguishable particles. The intrinsic entanglement of a many-body state is a basis-independent property[99], i.e., it is invariant under a unitary transformation over all modes on the particle creation/annihilation operators. In a few-body system, the Slater rank number or fermionic concurrence[99–101] can serve as basis-independent quantification of entanglement. In a many-body material system, it is impractical to measure these quantities, but one can construct different witnesses to set tight bounds among various entangled states and to fit observables accessible by solid-state measurements. A promising witness for indistinguishable fermions is operators sensitive to paired states[102] and constructed by at most two creation and two annihilation operators.

## Methods

### Time-resolved resonant inelastic X-ray scattering
The trRIXS is a photon-in-photon-out scattering process with a resonant intermediate state, whose cross-section reads as[76,89]

$$\mathcal{I}(q, \omega_s, \omega_i, t) = \frac{1}{2\pi N} \iiiint dt_1 dt_2 dt_1' dt_2' e^{i\omega_i(t_2-t_1)-i\omega_s(t_2'-t_1')}$$
$$\times g(t_1; t)g(t_2; t)l(t_1'-t_1)l(t_2'-t_2) \quad (9)$$
$$\times \langle \hat{\mathcal{D}}_{q_i\epsilon_i}^\dagger(t_2)\hat{\mathcal{D}}_{q_s\epsilon_s}(t_2')\hat{\mathcal{D}}_{q_s\epsilon_s}^\dagger(t_1')\hat{\mathcal{D}}_{q_i\epsilon_i}(t_1)\rangle$$

where $q = q_i - q_s$ ($\omega = \omega_i - \omega_s$) is the momentum (energy) transfer between incident and scattered photons, and $l(t) = e^{-t/\tau_{core}}\theta(\tau)$ the core-hole decay lifetime. For a direct transition, the dipole operator reads as

$$\mathcal{D}_{q\varepsilon} = \sum_{i\alpha\sigma} e^{-iq\cdot r_i}(M_{\alpha\varepsilon}c_{i\sigma}^\dagger p_{i\alpha\sigma} + h.c.), \quad (10)$$

where $c_{i\sigma}^\dagger$ ($c_{i\sigma}$) and $p_{i\alpha\sigma}^\dagger$ ($p_{i\alpha\sigma}$) denote the creation (annihilation) operators for valence and core-level electrons at site $i$ with spin $\sigma = \uparrow, \downarrow$. Since the transition-metal $L$-edge usually involves multiple core-level $p$ orbitals, we label them by $\alpha$, for which $M_{\alpha\varepsilon}$ is the matrix element of the dipole transition between each core level and the valence band via an $\varepsilon$-polarized photon. For a transition-metal $L$-edge trRIXS, the full derivation of the matrix elements is reported in ref. 76. While these dipole transitions preserve the total spin, the pair of photon absorption and emission events, described by $\hat{\mathcal{D}}_{q_s\epsilon_s}^\dagger(t_1')\hat{\mathcal{D}}_{q_i\epsilon_i}(t_1)$ in Eq. (9), may flip a spin due to the spin-orbit coupling of the core levels. This spin-flip process is maximized for the $\pi - \sigma$ polarization and, therefore, provides a good estimate of the dynamical structure

factor[75,77,103–105]. Throughout this paper, we exclusively employ such a polarization configuration.

### Extended Hubbard model
The extended Hubbard model Hamiltonian reads as

$$\mathcal{H} = -t_h \sum_{i\sigma}\left[c_{i\sigma}^\dagger c_{i+1,\sigma} + H.c\right] + U \sum_i n_{i\uparrow}n_{i\downarrow}$$
$$+ V \sum_{i,\sigma,\sigma'} n_{i\sigma}n_{i+1,\sigma'} + \mathcal{H}_{core}, \quad (11)$$

where $c_{i\sigma}$ ($c_{i\sigma}^\dagger$) annihilates (creates) a valence electron and $n_{i\sigma} = c_{i\sigma}^\dagger c_{i\sigma}$ is the number operator. The valence electrons form a single band with nearest-neighbor hopping amplitude $t_h$, on-site Coulomb repulsion $U$, and nearest-neighbor interaction $V$. Although the formalism is general, we choose here model parameters capturing the physics of cuprate chain compounds such as $Ba_{2-x}Sr_xCuO_{3+\delta}$, namely $U = 8t_h$ and $V = -t_h$[79,89]. The ground state of EHM with these parameters at quarter filling was suggested to reside in proximity to a spin-triplet superconducting state[84,85,94–97].

To account for the x-ray absorption and emission processes, the full Hamiltonian in Eq. (11) also contains terms involving the core holes

$$\mathcal{H}_{core} = \sum_{i\alpha\sigma} E_{edge}(1 - n_{i\alpha\sigma}^{(p)}) - U_c \sum_{i\alpha\sigma\sigma'} n_{i\sigma}(1 - n_{i\alpha\sigma'}^{(p)})$$
$$+ \lambda \sum_{\substack{i\alpha\alpha'\\\sigma\sigma'}} p_{i\alpha\sigma}^\dagger \chi_{\alpha\alpha'}^{\sigma\sigma'} p_{i\alpha'\sigma'}. \quad (12)$$

Here, $p_{i\alpha\sigma}$ ($p_{i\alpha\sigma}^\dagger$) annihilates (creates) a core-level electron with multiple degenerate orbitals labeled by $\alpha$, corresponding to the $2p_{x,y,z}$ orbitals in transition-metal $L$-edge RIXS, and $n_{i\alpha\sigma}^{(p)} = p_{i\alpha\sigma}^\dagger p_{i\alpha\sigma}$ is the core-level electronic number operator. The potential $U_c$ describes the attractive interaction between the core-hole holes and valence-level electrons and is fixed at $4t_h$ for all the $2p$ orbitals[77,87,106]. The edge energy $E_{edge}$ is selected as 938 eV to represent the Cu $L$-edge x-ray absorption and the spin-orbit coupling $\lambda$ of the core states is set to 13 eV[106,107].

## Data availability
The numerical data that support the findings of this study are available from the corresponding authors upon reasonable request. The data generated in this study are provided in the figshare repository.

## Code availability
The relevant scripts to reproduce all figures are available at the figshare repository. Other codes are available from the corresponding authors upon reasonable request.

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

## Acknowledgements

We acknowledge insightful discussions with R.C. de Almeida, S. Ding, P. Hauke, and P. Laurell. J.H. and Y.W. acknowledge support from U.S. Department of Energy, Office of Science, Basic Energy Sciences, under Early Career Award No. DE-SC0022874. D.R.B. and M.M. are primarily supported by the U.S. Department of Energy, Office of Science, Basic Energy Sciences, under Early Career Award No. DE-SC0022883. D.R.B. also acknowledges funding by the Swiss National Science Foundation through Project No. P400P2_194343. T.L. and M.L. acknowledge the support from U.S. Department of Energy (DOE), Office of Science, Basic Energy Sciences (BES), award No. DE-SC0020148. This research used resources of the National Energy Research Scientific Computing Center (NERSC), a U.S. Department of Energy Office of Science User Facility located at Lawrence Berkeley National Laboratory, operated under Contract No. DE-AC02-05CH11231 using NERSC award BES-ERCAP0020159.

## Author contributions

Y.W. and M.M. conceived the project. J.H., U.B, D.R.B., and Y.W. performed the calculations and data analysis. J.H., U.B., T.L., D.R.B., M.L., M.M., and Y.W. wrote the manuscript.

## Competing interests

The authors declare no competing interests.
