## [Peer Review File · Nature Communications]

REVIEWER COMMENTS

Reviewer #1 (Remarks to the Author):

Hales et al. propose an approach to witnessing the entanglement dynamics in quantum materials from time-resolved resonant inelastic x-ray scattering (trRIXS) using a time-dependent formulation of the quantum Fisher information (QFI). QFI, a witness of multipartite entanglement, has previously been rigorously related to equilibrium spectra and measured using inelastic neutron scattering. An extension to non-equilibrium spectroscopic experiments is highly desirable, in part because it could help characterize exotic transient states, and in part because, e.g., thermalization processes are often understood in entanglement terms. The authors thus set out to tackle a clear and highly important challenge: how to measure and control the time-dependent entanglement in macroscopic light-driven systems. They apply their approach to the 1D extended Hubbard model, relevant to cuprate chains. I note here that most of the entanglement witness literature has focused on two-level systems (e.g. qubits, spin-1/2 systems and spinless fermions), which tend to be easier to handle. However, extensions to spinful fermions are welcome, in order to support analysis of more general correlated electron systems.

That said, I do not recommend publication in Nature Communications. The first reason is that I have concerns, outlined below, about their QFI approach being less generally applicable than claimed in the manuscript. The second reason is that the later part of the manuscript, which focuses on applying the approach to the extended Hubbard chain and the possibility of enhancing its entanglement using light, would benefit from a somewhat deeper study comparing QFI to other entanglement measures (perhaps entanglement entropy dynamics has already been studied?). Otherwise it serves primarily as an example, and would not seem to significantly advance understanding of the system in its own right.

The authors develop a formalism to relate the transient structure factor $S(q, \omega, t)$ measured in trRIXS pump-probe experiments to the QFI density convolved with the probe envelope function. There appears to be two important limitations to their derivations. First, the definition for the QFI density in Eq. (1) assumes a pure state. Of course, actual materials tend to be in mixed thermal states. In the equilibrium case, the finite-T QFI tends to be simply reduced compared to its T=0 value. Does the same trend apply out-of-equilibrium? Is it justified to assume the initial state to be pure in an trRIXS experiment? Certainly many regular RIXS experiments are conducted at temperatures comparable to or larger than the energy scale of magnetic excitations. If it is justified to consider the initial state as pure, is it also justified to treat the state of the system after subject to the pump as pure? By unitarily time evolving an initially pure state the authors ensure these assumptions hold for

the finite-size Hubbard system they calculate QFI for, but do they hold (approximately) also under experimental conditions?

Second, in the derivation given in Sec. I of the supplementary information it is stated that the QFI is defined as $f_Q = 4 \langle \hat{\rho}_{-q}^s(t) \hat{\rho}_q^s(t) \rangle / N$. This form of the QFI is then used to derive Eqs. (4) and (5) of the main text, which are the main results of the manuscript. But doesn't this f_Q expression correspond to only the first term in Eq. (1) of the main text? If so, is there an assumption that the cross terms in the connected correlation function simply vanish? Although they likely vanish in the 12-site chain studied numerically in the paper, that assumption is not justified in general. It may be broken, for example in spontaneously symmetry-broken states or in the presence of polarizing fields. Crucially, this would imply that the self-consistent QFI calculated from Eq. (5) cannot be generally used to bound the entanglement depth. Non-vanishing cross term may also exacerbate the deviation between the exact and self-consistent lines in Fig. 5D.

If these limitations are indeed present it needs to be stated clearly in the manuscript.

Additional questions and comments:

1. In SI Eq. (4) has there been a variable substitution $x=t-\tau$? If SI Eq. (4) is correct, then the definitions of C_m given after Eq. (5) of the main text or SI Eq. (5) appear to be incorrect. In SI Eq. (4) the prefactor $1/(8\pi\sigma_{pr}^2)$ is outside the sum, so its inverse should not be appear in C_m . Also please double check the m -dependence of the exponent in C_m , as well as the sign between the integral over $S(q,\omega,t)$ and the sum Eq. (5).

2. The introduction mentions that the complexity of entangled wavefunctions scales exponentially with the system size. The statement could be made more precise as to which notion of complexity it refers to. For example, the complexity of a tensor network representation can be reduced below that of the diagonalization problem.

3. It is implied that the quantum Cramér-Rao bound is not well-defined for general fermionic systems. The phrase "the quantum Cramér-Rao bound" would typically refer to the bound on parameter estimation from quantum metrology, which, as far as I am aware is well-defined for any

quantum system. The authors appear to use the term as synonymous with bounds on the entanglement depth from QFI, but this is non-standard usage.

4. The notation in Sec. II B can be understood through referencing the methods, but you could consider making it more accessible to non-RIXS-specialists.

5. In Sec. II C the momentum $q=\pi/6$ is chosen for the evaluation of QFI based on it exhibiting the most evident spectral changes and capturing the longest-range correlations supported by the system size. Is there a physical argument for why this momentum is particularly interesting in the model? In equilibrium, the most interesting momentum choice is typically related to an order parameter.

6. Sec. II D: Why is Ref. [94] cited in the context of double occupation? It concerns QFI after a quench in an Ising model. Towards the end of the section, isn't the spin-mode entanglement probed here "associated with fermions" since the model is fermionic?

7. SI Eq. (1) appears to miss a $1/\sigma_{pr}^2$ factor and the sign in the exponent of $e^{i\omega \tau}$ may be wrong. Do the authors really Taylor expand in τ to obtain SI Eq. (3)? It seems sufficient to use the standard delta function identity $\int d\omega e^{-i\omega \tau} = 2\pi \delta(\tau)$.

8. A minor typo: 'self-consistant' in Fig. 5D caption.

Reviewer #2 (Remarks to the Author):

In the manuscript titled "Witnessing Light-Driven Entanglement using Time-Resolved Resonant Inelastic X-Ray Scattering", the authors have proposed time dependent quantum Fisher information (t-QFI) as a witness to detect and quantify multipartite entanglement in driven systems through time-resolved resonant inelastic x-ray scattering (trRIXS). The manuscript is well written and can be accepted for publication after the following queries are answered:

1. The time dependent quantum Fisher information $fQ(q,t)$ is obtained from equation (5) self-consistently where the derivative terms of fQ differentiate the static QFI from the time dependent QFI. The structure factor $S(q,\Omega,t)$ is found from equation (6). However, it is not very clear to me how the additional time-derivative fQ term is accessed experimentally. The authors calculate QFI's

neglecting this additional term and call it “snapshot QFI”. However, it is found that the snapshot QFI is insufficient to capture the true t-QFI.

2. Even though $S(q, \Omega, t)$ is found from equation (6), the authors clearly state that a tRIX spectrum is not identical to spin structure factor $S(q, \Omega, t)$. In such a case, how valid is their t-QFI estimation from a trRIXS experiment?

3. The authors envisage witnessing a m-particle entangled state through t-QFI but the example that they have used shows only a bipartite entangled state. Is this a drawback of the specific example that they have used or a shortcoming of the technique itself? Please clarify.

4. Finally, I am not very sure of the reason for choosing their system of demonstration as a one-dimensional extended Hubbard model. It is true that it's a correlated system but multipartite entanglement in this system has not been probed in great details. If it has been done, please give suitable references. Since the authors propose t-QFI as a new technique to probe time-dependent multipartite entanglement, it would be nice if the authors show the limits in which the t-QFI reduced to the static QFI limit in their extended Hubbard model. Would it be possible for the authors to compare systems like low dimensional magnets where multipartite entanglement has been shown to exist on a firm footing?

Reply to Reviewer #1

We thank the Reviewer for carefully reading our work and for the positive assessment of both our methodology and results. The Reviewer states that “*The authors thus set out to tackle a clear and highly important challenge: how to measure and control the time-dependent entanglement in macroscopic light-driven systems.*”. He/She also points out that our study has pushed the field forward, as “*most of the entanglement witness literature has focused on two-level systems*” and “*an extension to non-equilibrium spectroscopic experiments is highly desirable*”. The Reviewer also raises insightful suggestions and comments, which helped us improve the manuscript.

Below we address the questions raised by the Reviewer. We indicate in **blue** the original reports, in **black** our reply, and in **red** our changes in the revised manuscript.

(1) Reviewer’s question: “The first reason is that I have concerns, outlined below, about their QFI approach being less generally applicable than claimed in the manuscript. The second reason is that the later part of the manuscript, which focuses on applying the approach to the extended Hubbard chain and the possibility of enhancing its entanglement using light, would benefit from a somewhat deeper study comparing QFI to other entanglement measures (perhaps entanglement entropy dynamics has already been studied?). Otherwise it serves primarily as an example, and would not seem to significantly advance understanding of the system in its own right.”

Our Reply: We thank the Reviewer for the opportunity to clarify our message. The Reviewer raises two distinct concerns, namely that (1) the quantum Fisher information (QFI) approach investigated here is not general, and (2) there is a lack of comparison with other measures of entanglement.

First, we would like to stress that our work builds upon a general result for equilibrium systems by Hauke *et al.* (*Nat. Phys.* **12**, 778), which rigorously connects the QFI for local operators with the equilibrium spectral response functions. Its extension to the nonequilibrium structure factor in Sec. II.A of the main text is as general as the Hauke’s conclusions, and only makes use of the spin operators as a concrete experimental example and a typical choice in quantum information. We then identify a model system where it is possible to observe a light-driven enhancement of many-body entanglement and focus on validating our protocol for future nonequilibrium spectroscopy experiments.

About the second concern, we stress that the main motivation for our work is to explore an entanglement witness that is rigorously linked to experimentally accessible spectroscopic observables (such as the trRIXS spectrum) and thus guide future experimental work. Such a firm relation with the dynamical susceptibility exists, so far, only for the QFI, and this is the primary reason for focusing the narrative on this quantity. Following the Reviewer’s suggestion, we calculated the dynamical evolution of the entanglement entropy (EE) using the same model (a quarter-filled EHM with $U = 8t_h$ and $V = -t_h$) and pump-probe parameters (amplitude $A_0 = 1$, frequency $\Omega = 10t_h$, and probe width $\sigma_{\text{pump}} = 3t_h^{-1}$) of Fig. 4 (main-text). Figure R1, included as Fig. S4 of the Supplementary Information, compares the nonequilibrium evolution of EE S_{ent} and QFI f_Q . As visible in the figure, EE and QFI are both enhanced by the laser pulse within the same timescale and exhibit a similar evolution: a slow, monotonic increase until the middle of the pulse ($t = 0$, when the pump field

Fig. R1: Comparison between transient entanglement entropy and QFI density for a light-driven extended-Hubbard model. Model parameters ($U = 8t_h$ and $V = -t_h$) and pump conditions ($A_0 = 1$ and $\Omega = 10t_h$) are the same as those in Fig. 4 of the main text.

peaks), followed by a saturation accompanied by slow oscillations that persist long after the pump excitation. The consistency between QFI and EE dynamics further indicates that the transient QFI correctly captures the enhanced entanglement of the quantum state.

Changes made: We have calculated the entanglement entropy of the light-driven extended Hubbard model and compared it with the QFI dynamics in the revised manuscript. This part is included in the Supplementary Information as the “Sec. IV. Nonequilibrium Entanglement Entropy”.

(2) Reviewer’s question: “The authors develop a formalism to relate the transient structure factor $S(q, \omega, t)$ measured in trRIXS pump-probe experiments to the QFI density convolved with the probe envelope function. There appears to be two important limitations to their derivations. First, the definition for the QFI density in Eq. (1) assumes a pure state. Of course, actual materials tend to be in mixed thermal states. In the equilibrium case, the finite- T QFI tends to be simply reduced compared to its $T = 0$ value. Does the same trend apply out-of-equilibrium? Is it justified to assume the initial state to be pure in an trRIXS experiment? Certainly many regular RIXS experiments are conducted at temperatures comparable to or larger than the energy scale of magnetic excitations. If it is justified to consider the initial state as pure, is it also justified to treat the state of the system after subject to the pump as pure? By unitarily time evolving an initially pure state the authors ensure these assumptions hold for the finite-size Hubbard system they calculate QFI for, but do they hold (approximately) also under experimental conditions?”

Our response: We thank the Reviewer for the opportunity to clarify our message. Here, the Reviewer is concerned about the robustness of the observed trends in the realistic case of a thermal initial state. As we discuss below, our conclusions apply equally well to pure and mixed states out of equilibrium, including experimental regimes relevant to the study of quantum materials.

First, we stress that the QFI can be extracted from the trRIXS cross-section both in

the case of pure and mixed initial states. While our numerical simulations are performed using a pure initial state, the self-consistent equations reported in Eqs. (4) and (5) are also valid for a thermal ensemble. The quantum average notation $\langle \dots \rangle$ in any instantaneous operator [e.g., $f_Q(q, t)$] and pump-probe spectral observables [e.g., the dynamical structure factor $S(q, \omega, t)$ or trRIXS intensity $I(q, \omega, t)$] can straightforwardly be extended to describe a thermal ensemble $\text{Tr}[e^{-\beta\mathcal{H}}/\mathcal{Z} \dots]$ by incorporating the corresponding Boltzmann weights. Since Eq. (5) is linear in $S(q, \omega, t)$ and f_Q , the same applies to the trRIXS cross-section and the QFI density.

Next, [redacted]

Finally, we justify the unitary evolution adopted in our work. This assumption implies that the Hamiltonian describes a closed quantum system including the laser field together with valence and core-level electrons. This closed-system assumption is well-justified in the ultrafast dynamics limit, where the lattice degrees of freedom adjust very slowly. The latter timescale is dictated by the phonon frequencies. Taking the 1D cuprate chain simulated in this paper as an example, the energy of one of the most strongly coupled phonons is $\sim 0.1t_h$ (~ 70 meV in laboratory units), which corresponds to a timescale of $2\pi/0.1t_h \approx 60t_h^{-1}$. However, the light-driven entanglement dynamics stemming from pure electronic correlations already occur within a timescale of $\sim 5t_h^{-1}$. Furthermore, as shown in related work by some of the authors [arXiv:2209.02081, Phys. Rev. Lett. (2023), in press], coupling to an external bath (treated semi-classically via the Lindblad master equation) qualitatively preserves the nonequilibrium QFI dynamics and departs from the unitary evolution only for large coupling constants and at a much longer timescale.

Changes made:

We have modified the definition and self-consistent formulation of the nonequilibrium dynamical structure factor to explicitly account for the thermal ensemble average. After Eq. (1) of the SI, we have added the following text “Here, the average notation is not restricted to a pure state, but can also be generalized to a thermal ensemble

$$\begin{aligned} & \langle \hat{O}_1(t_1) \hat{O}_2(t_2) \rangle \\ &= \text{Tr} \left[\frac{e^{-\beta\mathcal{H}}}{\mathcal{Z}} \hat{U}(-\infty, t_1) \hat{O}_1 \hat{U}(t_1, t_2) \hat{O}_2 \hat{U}(t_2, -\infty) \right]. \end{aligned} \quad (1)$$

The $\hat{U}(t_1, t_2)$ is the unitary time evolution operator, \mathcal{Z} is the partition function of the equilibrium state (at $t = -\infty$), and β is the inverse temperature.”

(3) Reviewer’s question: “Second, in the derivation given in Sec. I of the supplementary information it is stated that the QFI is defined as $f_Q = 4\langle \hat{\rho}_{-q}^s(t) \hat{\rho}_q^s(t) \rangle / N$. This form of the QFI is then used to derive Eqs. (4) and (5) of the main text, which are the main results of the manuscript. But doesn’t this f_Q expression correspond to only the first term in Eq. (1) of the main text? If so, is there an assumption that the cross terms in the connected correlation function simply vanish? Although they likely vanish in the 12-site chain studied numerically in the paper, that assumption is not justified in general. It may be broken, for example in spontaneously symmetry-broken states or in the presence of polarizing fields. Crucially, this would imply that the self-consistent QFI calculated from Eq. (5) cannot be generally used to

bound the entanglement depth. Non-vanishing cross term may also exacerbate the deviation between the exact and self-consistent lines in Fig. 5D. If these limitations are indeed present it needs to be stated clearly in the manuscript.”

Our response: We thank the Reviewer for raising this question. In our work, we calculate the expectation values of the full QFI expression, including both the connected and disconnected parts. However, since our 1D system does not exhibit spontaneous symmetry breaking, the disconnected part of the QFI density vanishes. Hence, in our manuscript the two expressions are equivalent.

More generally, our method is also applicable in presence of long-range order, as one can separate connected and disconnected components of the correlation function. The first part of the QFI operator corresponds to a full frequency integration of the dynamical structure factor yielding the sum of disconnected and connected parts of the correlation function. The disconnected part along is instead a separable spectral component corresponding to the elastic scattering peak centered at $\omega = 0$, which can be usually removed from the data (e.g., by fitting and subtraction). Once the elastic peak is subtracted, the remaining part of the spectrum can be mapped to the QFI through our self-consistent formula.

Changes made: We now discuss the disconnected part of the QFI and the presence of elastic peaks in the main text after Eq. (4): “**Note that we have assumed the absence of long-range magnetic order at the specific momentum q , which is the case for the simulations in this paper. If a long-range order is present, one should further subtract the elastic peak from the structure factor, whose intensity corresponds to the disconnected part (second term) in Eq. (1).**”

(4) Reviewer’s question: “In SI Eq. (4) has there been a variable substitution $x = t - \tau$? If SI Eq. (4) is correct, then the definitions of C_m given after Eq. (5) of the main text or SI Eq. (5) appear to be incorrect. In SI Eq. (4) the prefactor $1/(8\pi\sigma_p r^2)$ is outside the sum, so its inverse should not be appear in C_m . Also please double check the m-dependence of the exponent in C_m , as well as the sign between the integral over $S(q, \omega, t)$ and the sum Eq. (5).”

Our response: We thank the Reviewer for pointing this out. Indeed, we introduce the substitution $x = \tau - t$ in SI Eq.(4) and have now added a line to clarify our message. Further, the correct expression for C_m should read $C_m = -(1/\sigma_{pr}\sqrt{\pi}) \int_{-\infty}^{\infty} e^{-x^2/\sigma_{pr}^2} x^{2m} dx = -(\sigma_{pr}^{m-1/2}/\sqrt{\pi})\Gamma(m + 1/2)$. Note that we have absorbed the minus sign in this definition. We have also confirmed that this is exclusively a typo in the manuscript and not an error in our code.

Changes made: To clarify our derivation, we **have added a line about $x = \tau - t$ just above SI Eq. (5) and corrected the expression of C_m just below SI Eq. (6) and in Eq. (5) of the main text.** [Note that the original Eqs. (4) and (5) become the Eqs. (5) and (6) in the revised manuscript.]

(5) Reviewer’s question: “The introduction mentions that the complexity of entangled wavefunctions scales exponentially with the system size. The statement could be made more precise as to which notion of complexity it refers to. For example, the complexity of a tensor network representation can be reduced below that of the diagonalization problem.”

Our response: We thank the Reviewer for pointing out the need to sharpen our introduction. Here, we refer to the wavefunction complexity from the point of view of a measurement of Rényi entropy and of multi-point correlations using cold-atom approaches (e.g., the quantum gas microscope). Since these methods are based on sampling over Fock states, a reliable measurement of these quantities requires sufficient averaging over the Hilbert space. Both the sampling number and the Hilbert space scale exponentially with the system size. In a general quantum system, a full tensor representation of the wavefunction has the same degrees of freedom as the Hilbert space, which scales as $O(e^N)$. For specific systems (e.g. low-entangled systems with local interactions), an adaptive truncation of this dimension is instead possible. This truncation is usually referred to as the bond dimension in tensor representation and can be used as an entanglement metric for the many-body wavefunction [see e.g., Sci. Rep. 6, 30188 and Phys. Rev. Lett. 123, 170504]. However, we note that the bond dimension is not an experimentally accessible quantity.

Changes made: At the Reviewer’s suggestion, we clarified the notion of complexity in the introduction of the revised manuscript. The revised text now reads as “Entangled wavefunctions in synthetic few-body quantum simulators can be experimentally characterized [15,17-20] through the Rényi entropy [21-25] and multi-point correlations [26-33]. **However, the complexity of their measurement increases with the Hilbert-space dimension and scales exponentially with the system size. As solid-state measurements are restricted to a limited number of macroscopic observables, a tomography of electronic wavefunctions in quantum materials becomes impractical [34-37].**”

(6) Reviewer’s question: “It is implied that the quantum Cramér-Rao bound is not well-defined for general fermionic systems. The phrase ‘the quantum Cramér-Rao bound’ would typically refer to the bound on parameter estimation from quantum metrology, which, as far as I am aware is well-defined for any quantum system. The authors appear to use the term as synonymous with bounds on the entanglement depth from QFI, but this is non-standard usage.”

Our response: We thank the Reviewer for pointing out an inaccuracy of the previous version of our manuscript. Indeed, the quantum Cramér-Rao bound, i.e., $(\Delta\theta)^2 \geq 1/mF_Q[\rho, H]$, is a well-defined metrology concept for any quantum system regardless of the specific model. In our first submission, we meant to refer to the upper boundary of the QFI expectation value for any k -producible state. In spin qubits, combining the QFI upper bound at the maximally entangled state (the GHZ state) and the quantum Cramér-Rao bound, we can obtain the optimal bound for the parameter estimation. The quantum Cramér-Rao bound does not serve as synonym for the QFI upper bound. In fermionic systems, the connection between QFI of a local operator and the entanglement depth of the many-electron wavefunction is unclear, but the concept of a quantum Cramér-Rao bound still holds.

Changes made: At the Reviewer’s suggestion, we corrected the way we refer to the quantum Cramér-Rao bound in the introduction of the revised manuscript. The revised text now reads as “**While the connection between QFI and multiparticle entanglement is well established for spin operators in magnetic materials, analytical formulations for general fermionic systems are still lacking.**” We similarly corrected the last paragraph of Sec.II A: “**The self-consistently calculated $f_Q(q, t)$ serves the purpose of witnessing entanglement in a transient k -partite**

quantum state when exceeding its operator-specific boundary [50].”

(7) Reviewer’s question: “The notation in Sec. II B can be understood through referencing the methods, but you could consider making it more accessible to non-RIXS-specialists.”

Our response: We agree with the Reviewer that Sec.II B could be more accessible for non-RIXS experts. We have substantially revised this section to include necessary information so the readers do not have to refer to the Methods for a general understanding of trRIXS measurements.

Changes made: Following the Reviewer’s suggestion, we expanded the introduction about trRIXS in the first paragraph of Sec.II B: “As detailed in Methods, trRIXS is a photon-in-photon-out x-ray scattering process involving an intermediate state with finite lifetime. Due to the spin-orbit coupling at the core level (e.g., the $2p$ orbitals for the transition-metal L -edge RIXS), this intermediate state can involve spin flip events and couple to magnetic excitations of the valence band [77]. Therefore, trRIXS is sensitive to spin excitations and can be used to probe the nonequilibrium spin dynamics of light-driven materials [78].”

Moreover, we have added a brief discussion about the selection of polarization and the format of trRIXS spectra: “The trRIXS cross-section, denoted as $\mathcal{I}(q, \omega_i - \omega, \omega_i, t)$ in Methods, depends on energy, momentum, and polarization of both the incident and scattered photons. In our simulation, we select a scattering geometry with π -polarized (parallel to the scattering plane) incident photons and σ -polarized (perpendicular to the scattering plane) scattered photons, which maximizes the spin-flip contribution to the trRIXS cross-section [77,78]. Due to the focus on spin entanglement, we keep this polarization configuration fixed throughout the paper and omit the polarization subscripts in \mathcal{I} . The trRIXS spectrum comes with two energy axes for the incident photon energy ω_i and the energy loss ω (difference between incident and scattering photon energies).”

Finally, we have moved the definition of the trRIXS-related notations to the main text after Eq. (6): “Here, $\hat{D}_{q_i \epsilon_i}$ and $\hat{D}_{q_s \epsilon_s}^\dagger$ denote the dipole-transition operators for the incident and scattering process, respectively; $M^{(\text{in})}$ and $M^{(\text{out})}$ indicate the corresponding matrix elements and τ_{core} is the core-hole lifetime. The definition of these factors and the full expression of the trRIXS cross-section $\mathcal{I}(q, \omega_i - \omega, \omega_i, t)$ are explained in the Methods section.”

(8) Reviewer’s question: “In Sec. II C the momentum $q = \pi/6$ is chosen for the evaluation of QFI based on it exhibiting the most evident spectral changes and capturing the longest-range correlations supported by the system size. Is there a physical argument for why this momentum is particularly interesting in the model? In equilibrium, the most interesting momentum choice is typically related to an order parameter.”

Our response: We thank the Reviewer for the opportunity to clarify our choice. First, since our 1D system — heavily doped and with mixed (repulsive and attractive) interactions — does not exhibit a symmetry-breaking transition, there is no order parameter to follow, rather only fluctuating spin excitations. At such a doping level, where we can map the fermions into a slave-boson picture (see, e.g., Phys. Rev. B 99, 205102), the resulting spinon Fermi surface is extremely small and the low-energy spin excitations mainly occur at small momentum. In our Supplementary Information, we provide the QFI evolution for all the momenta allowed by our cluster. It is indeed the QFI at $q = \pi/6$ that exhibits the greatest

enhancement, while the QFI at another small momentum $q = \pi/3$ shows a more moderate enhancement. The momentum-dependent QFI enhancement of our 1D model validates our choice of $q = \pi/6$ for both the study of the spin fluctuations and of the transient QFI dynamics.

Changes made: To emphasize the choice of $q = \pi/6$, we have now revised the corresponding main text sentence as “When calculating the time-dependent QFI, we focus on a specific momentum, $q = \pi/6$. Distinct from undoped antiferromagnets previously employed to witness equilibrium entanglement [56], this small-momentum wavevector exhibits the most evident spectral changes and captures the longest-range correlations supported by our system size.” We have also added a sentence at the end of the same paragraph to further emphasize the absence of long-range order: “The 1D system simulated in this paper does not have a spontaneous symmetry breaking, which simplifies the entanglement analysis of the spin fluctuations.”

(9) Reviewer’s question: “Sec. II D: Why is Ref. [94] cited in the context of double occupation? It concerns QFI after a quench in an Ising model. Towards the end of the section, isn’t the spin-mode entanglement probed here “associated with fermions” since the model is fermionic?”

Our response: We thank the Reviewer for pointing out this clerical error (Ref. 94, Pappalardi *et al.* *J Stat. Mech.* 2017, 053104). We apologize for this mistake and have now corrected the reference, which should be Lorenzana *et al.* *Phys. Rev. B*, 72, 224511

Changes made: We have corrected the Ref. 94 into “Lorenzana *et al.* *Phys. Rev. B*, 72, 224511 (2005)”.

(10) Reviewer’s question: “SI Eq. (1) appears to miss a $1/\sigma_{\text{pr}}^2$ factor and the sign in the exponent of $e^{i\omega\tau}$ may be wrong. Do the authors really Taylor expand in τ to obtain SI Eq. (3)? It seems sufficient to use the standard delta function identity $\int d\omega e^{-i\omega\tau} = 2\pi\delta(\tau)$.”

Our response: We thank the Reviewer for bringing the pre-factor $1/\sigma_{\text{pr}}^2$ typo to our attention. We have added the pre-factor in SI Eq. (1) and confirmed that the normalization is correct in our simulations. To further clarify the exponent sign of $e^{i\omega\tau}$, in the main text Eq. (2) we have switched the dummy variables t_1 and t_2 and included a minus sign in front of q making the SI Eq. (1) and main text Eq. (2) consistent. Regarding the identity $\int d\omega e^{-i\omega\tau} = 2\pi\delta(\omega)$, we note that the integral in SI Eq. (1) is coupled in variables τ and $\bar{\tau}$ due to the $\langle \hat{\rho}_{-q}^s(\bar{\tau} + \tau/2) \hat{\rho}_q^s(\bar{\tau} - \tau/2) \rangle$ term, so the identity $\int d\omega e^{-i\omega\tau} = 2\pi\delta(\omega)$ is not applicable unless time-translational invariance is preserved (equilibrium system). The Taylor expansion with τ in SI Eq. (1) is introduced to decouple the integral that allows us to derive SI Eq. (3).

Changes made: We have corrected the prefactor in the Eq. (1) of SI, by adding the term $1/\sigma_{\text{pr}}^2$. We have also flipped dummy variables t_1, t_2 and added a minus sign in front of q in main text Eq. (2).

(11) Reviewer’s question: “A minor typo: ‘self-consistant’ in Fig. 5D caption.”

Our response: We thank the Reviewer for pointing out this typo, which has now been

fixed in our revised manuscript.

Changes made: We have edited the label in Fig. 5d to correctly read as “self-consistent”.

Reply to Reviewer #2

We thank the Reviewer for the careful reading of our manuscript. We are grateful for their positive evaluation of our work and are glad that they find “*The manuscript is well written and can be accepted for publication...*” Additionally, we address each of the Reviewer’s comments below (the comments are colored in **blue** and our revisions colored in **red**):

(1) Reviewer’s question: “The time dependent quantum Fisher information $f_Q(q, t)$ is obtained from equation (5) self-consistently where the derivative terms of f_Q differentiate the static QFI from the time dependent QFI. The structure factor $S(q, \omega, t)$ is found from equation (6). However, it is not very clear to me how the additional time-derivative f_Q term is accessed experimentally. The authors calculate QFI’s neglecting this additional term and call it “snapshot QFI”. However, it is found that the snapshot QFI is insufficient to capture the true t-QFI.”

Our response: The time-derivative terms can be experimentally measured by differentiating the time-dependent RIXS spectra along the pump-probe delay axis using a standard finite-difference method (FDM). Our simulations yield theoretical trRIXS data as close to the expected experimental signal as possible, and account for finite probe width, finite core-hole lifetime, and polarization-dependent matrix elements. Starting from these calculations, our QFI is extracted from the stroboscopic trRIXS signal along the pump-probe time delay axis as it would be done in an ultrafast x-ray scattering experiment.

As acknowledged by the Reviewer, a key finding of our work is that a single snapshot of trRIXS (at a certain time t) is insufficient to inform the transient QFI at the corresponding time and one must also account for its relation with multiple snapshots within a range of time delays. The additional high-order derivatives in Eq. (5) of the main text were obtained through the FDM and discrete Fourier transform of the entire sequence of trRIXS snapshots. We emphasize that, in order to keep our work as close as possible to a real experimental scenario, we did not rely on information which cannot be accessed in real trRIXS experiments (e.g., the two-point correlation functions, or the instantaneous wavefunctions).

Changes made: We have clarified our theoretical analysis in Sec. II.C: “We then proceed to evaluate the QFI as it would be done in real trRIXS experiments, i.e., by only assessing the sequence of trRIXS snapshots without additional theoretical knowledge about the state of the driven system.” In addition, we provide more details about the numerical evaluation of these high-order derivatives, by adding the following sentence to the main text: “We evaluate these high-order derivatives using finite difference methods, starting from the time sequence of trRIXS spectra, and then solving the self-consistent equations.”

(2) Reviewer’s question: “Even though $S(q, \omega, t)$ is found from equation (6), the authors clearly state that a trRIX spectrum is not identical to spin structure factor $S(q, \omega, t)$. In such a case, how valid is their t-QFI estimation from a trRIXS experiment?”

Our response: Indeed, trRIXS does not exactly correspond to $S(q, \omega, t)$. However, there exists a robust link between the RIXS spectrum and the dynamical spin structure factor for a large variety of experimentally relevant systems such as transition-metal oxides. This statement primarily applies to the spin excitation energy, while the absolute value of their spectral weight is usually less accurate. There are systematic approaches to correctly match the RIXS spectrum to $S(q, \omega)$, such as perturbatively adding contributions of higher-order

Fig. R2: Left: QFI dynamics evaluated using the simulated trRIXS intensity with large core-hole lifetime ($\tau_{\text{core}} = 1/2t_h$, red curve), compared with the exact QFI evolution obtained by instantaneous wavefunctions (black dashed curve). This panel is the same as Fig. 5b in the main text. Right: QFI dynamics evaluated using the corrected trRIXS spectra, which are scaled by an overall factor (1.163) determined by the ratio of equilibrium $S(q, \omega)$ and RIXS intensity. The scaled values match the exact results throughout the entire time evolution.

correlation functions [see Phys. Rev. X 6, 021020]. In the present work, for an extremely large τ_{core} , the absolute deviation between the trRIXS signal and the nonequilibrium $S(q, \omega, t)$ is no greater than that at equilibrium and the relative deviation ($\sim 10 - 15\%$) is even smaller. To demonstrate this, we rescaled the entire trRIXS spectrum for a large $\tau_{\text{core}} = 1/2t_h$ according to the mismatch between the RIXS intensity and $S(q, \omega)$ at equilibrium. Through our self-consistent iterations, the QFI dynamics extracted from the scaled trRIXS almost exactly matches the exact solutions, including the post-pump oscillations [see Fig. R2]. Such a correction is also applicable in real experiments, since the precise $S(q, \omega)$ at equilibrium can be measured using inelastic neutron scattering. Therefore, we conclude that trRIXS provides valuable experimental access to the instantaneous spin structure factor $S(q, \omega, t)$ and the transient QFI.

Changes made: We have included a discussion about the relation between the trRIXS intensity and the spin structure factor $S(q, \omega, t)$ for non-negligible core-hole lifetime: “When the lifetime is not negligible, the conversion of the equilibrium RIXS intensity into the $S(q, \omega)$ requires a more systematic approach. One method consists of calculating the four-particle response function as the lowest-order perturbative expansion of the lifetime τ_{core} [79]. For trRIXS, one can also correct for the finite lifetime effects by using an overall scaling factor determined by the equilibrium RIXS intensity and $S(q, \omega)$, both of which can be independently measured. As discussed in Sec. II of the SI, this correction provides a good approximation of the long-term dynamics even for a large core-hole lifetime.”

(3) Reviewer’s question: “The authors envisage witnessing a m-particle entangled state through t-QFI but the example that they have used shows only a bipartite entangled state. Is this a drawback of the specific example that they have used or a shortcoming of the technique itself? Please clarify.”

Our response: Driving an experiment-relevant system to a more entangled state is one of ultimate goals in the study of light-driven quantum materials. Achieving this objective simultaneously requires (a) *proving the ability to witness entanglement* in a nonequilibrium

setting and (b) *discovering light-induced/enhanced entanglement* in an experiment-relevant system. Our paper tackles both challenges on an equal footing. Regarding (a), we show that trRIXS can resolve the time-dependent QFI in a nonequilibrium system via a self-consistent extraction procedure. For (b), we find that the extended Hubbard model (as experimentally realized in certain low-dimensional cuprates) is a promising avenue to observe light-driven nonequilibrium entanglement at the crossover between a spin-dominant (Luttinger) state and a spin-triplet superconducting state. The observation of bipartite entanglement is a feature of this specific example and we cannot exclude that m -particle entangled states might be discovered in future experiments on different systems.

It is worth mentioning a recent work pointing out another promising direction. Suresh *et al.* proposed a spintronic device with two ferromagnets separated by a normal metal [arXiv:2210.06634]. By injecting current into this device, their simulation suggested a substantial increase of entanglement (witnessed by the QFI density $f_Q > 3$). This paper acknowledges our present manuscript and discusses that trRIXS may be the only approach to test this nonequilibrium entanglement enhancement. Beyond laser-driven quantum materials, spintronic devices with current injections might be an intriguing new platform for testing the ideas described in our paper.

Changes made: We now cite this recent arXiv paper about spintronic systems, where higher entanglement was achieved by current injection, in the second-to-last paragraph of the Discussion section: “Additionally, since the nonequilibrium control of entanglement was recently discussed also in the context of spintronic devices [100], our self-consistent trRIXS approach could eventually be applied to probe time-dependent entanglement in non-laser-driven systems.”

(4) Reviewer’s question: “Finally, I am not very sure of the reason for choosing their system of demonstration as a one-dimensional extended Hubbard model. It is true that it’s a correlated system but multipartite entanglement in this system has not been probed in great details. If it has been done, please give suitable references. Since the authors propose t-QFI as a new technique to probe time-dependent multipartite entanglement, it would be nice if the authors show the limits in which the t-QFI reduced to the static QFI limit in their extended Hubbard model. Would it be possible for the authors to compare systems like low dimensional magnets where multipartite entanglement has been shown to exist on a firm footing?”

Our response: In this work, we chose to investigate the extended Hubbard model for two reasons. This model is (a) realized in systems under investigation with RIXS and trRIXS experiments, and (b) exhibits an out-of-equilibrium enhancement of its multipartite entanglement.

The 1D extended Hubbard model is a natural description of several low-dimensional transition-metal oxides, particularly 1D cuprates (*Science* 373, 1235 and *Phys. Rev. Lett.* 127, 197003). These materials exhibit clear signatures of spin-charge separation and orbital excitations (*Nature* 485, 82), and become superconducting upon doping (*PNAS* 117, 4565). These features make them an ideal platform to investigate the interplay of correlations and dimensionality in superconducting copper oxides, as well as to understand nonequilibrium states induced by light in these materials. These 1D or quasi-1D cuprates (represented by Sr_2CuO_3) are well characterized with RIXS at equilibrium (see, e.g. *Nature* 485, 82 and

Fig. R3: Evolution of the QFI density f_Q for a light-driven half-filled Hubbard model while varying **a** pump amplitude (with $\Omega = 10t_h$) and **b** pump frequency (with $A_0 = 1$). The QFI is evaluated at the nesting wavevector $q = \pi$. The boundaries are indicated by the dashed lines and corresponding entanglement depths are indicated in the right bar.

Nat. Commun. **9**, 5394) and provide a natural candidate for upcoming trRIXS experiments. [redacted] While the Reviewer is correct in stating that the multipartite entanglement in this material has not yet been probed in detail, this system represents a timely, and most immediate experimental link between RIXS and previous work with inelastic neutron scattering on quasi-1D spin chains (see *Phys. Rev. Lett.* **127**, 037201 and *Phys. Rev. B* **103**, 224434).

Besides connecting RIXS and inelastic neutron scattering, our 1D extended Hubbard model also exhibits a striking phase diagram enabling the detection of light-enhanced entanglement. By conducting rigorous DMRG simulations, some of the authors have recently demonstrated the existence of a triplet superconducting phase induced by the presence of near-neighbor attractive interactions (*Comm. Phys.* **5**, 257). The boundary of the triplet phase is closest to 1D cuprate's interaction parameters ($U = 8t_h$ and $V = -t_h$) at quarter filling. The proximity to this quantum phase transition boundary provides a unique opportunity to experimentally observe light-enhanced entanglement, and not just a randomization of degrees of freedom due to incoherent light-matter interaction processes. Since multipartite entanglement is maximized near quantum phase transitions, a light-manipulation of the effective electronic interactions (*Phys. Rev. Lett.* **115**, 187401 and *Phys. Rev. X* **12**, 011013) could nudge the system towards criticality, thus increasing the witnessed multipartite entanglement.

In addition, we would like to emphasize that the aforementioned mechanism is only expected in the extended Hubbard model, and not in the pure Hubbard model (see Fig. 6 of the main text). As the Reviewer mentioned, simple 1D spin chains and undoped materials, such as the one investigated with inelastic neutron scattering would provide a highly-entangled equilibrium ground state. However, this highly entangled ground state does not guarantee a light-induced enhancement of multipartite entanglement. As shown in Fig. R3, by varying the pump field the multipartite entanglement in the spin sector of the half-filled Hubbard model is always suppressed. This implies that directly driving the half-filled spin chain out of equilibrium does not help in building entanglement correlations and, as demonstrated in *Commun. Phys.* **4**, 212, that the build-up of multipartite spin entanglement in presence

of long-range (or quasi-long-range) orders is not as efficient as in systems with short-range orders/fluctuations.

To summarize, we chose to investigate the nonequilibrium entanglement dynamics in the extended-Hubbard model as we anticipate material realizations of this system to provide the very first proof-of-principle of light-enhanced entanglement in quantum materials and we regard this choice timely and well-matched to the current experimental landscape of the field.

Changes made: To motivate the choice of the extended-Hubbard model and highlight its relevance to experiments, we have revised the first paragraph of Sec. IIC “*TrRIXS and QFI in a Driven Extended Hubbard Model*”. Now the revised text reads as: “Given prior equilibrium RIXS experiments [80], we consider 1D cuprate chains as an ideal platform for the experimental verification of our results. Recent experiments in $\text{Ba}_{2-x}\text{Sr}_x\text{CuO}_{3+\delta}$ have identified the EHM with mixed-sign interactions [72] as their underlying model Hamiltonian (see Methods) and here we investigate its light-driven dynamics. Throughout this paper, we set the on-site (U) and the nearest-neighbor (V) interactions to $U = 8t_h$ and $V = -t_h$, respectively, corresponding to the characteristic values for cuprate chains [72,81]. The existence of this nearest-neighbor term V is crucial for the presence of many-body entanglement, as we discuss in Sec. IID.” In addition, we have emphasized the importance of this model, against the usually considered Hubbard model, at the end of Sec. IID: “In other words, the nearest-neighbor interaction, whose feature was recently identified in 1D cuprate chains but the phonon-mediated mechanism widely exists in transition-metal oxides, is crucial role towards achieving light control of entanglement in correlated materials.”

Summary of Changes

All revisions as recommended by the Reviewers have been marked **red** in the resubmitted manuscript. Here we summarize the changes:

1. In the 2nd paragraph on Page 1 we clarify the notion of complexity: “Entangled wavefunctions in synthetic few-body quantum simulators can be experimentally characterized [15,17-20] through the Rényi entropy [21-25] and multi-point correlations [26-33]. However, the complexity of their measurement increases with the Hilbert-space dimension and scales exponentially with the system size. As solid-state measurements are restricted to a limited number of macroscopic observables, a tomography of electronic wavefunctions in quantum materials becomes impractical [34-37].”

2. In the 1st paragraph on page 2 we correct the way we refer to the quantum Cramér-Rao bound: “While the connection between QFI and multiparticle entanglement is well established for spin operators in magnetic materials, analytical formulations for general fermionic systems are still lacking.”

3. In the 2nd paragraph of Sec. II A (page 2) we flipped the dummy variables in Eq. (2).

4. In the 2nd paragraph on page 3 (after Eq. (4)) we addressed the disconnected part of the QFI and the presence of elastic peaks: “Note that we have assumed the absence of long-range magnetic order at the specific momentum q , which is the case for the simulations in this paper. If a long-range order is present, one should further subtract the elastic peak from the structure factor, whose intensity corresponds to the disconnected part (second term) of Eq. (1).”

5. In the 2nd paragraph on page 3 (after Eq. (5)), we fixed the expression for C_m : “ $C_m = -(1/\sigma_{\text{pr}}\sqrt{\pi}) \int_{-\infty}^{\infty} e^{-x^2/\sigma_{\text{pr}}^2} x^{2m} dx = -(\sigma_{\text{pr}}^{m-1/2}/\sqrt{\pi})\Gamma(m+1/2)$ ”.

6. In the last paragraph of Sec.II A on page 3, we correct the way we refer to the quantum Cramér-Rao bound: “The self-consistently calculated $f_Q(q, t)$ serves the purpose of witnessing entanglement in a transient k -partite quantum state when exceeding its operator-specific boundary [50].”

7. In the beginning of Sec.II B (page 3) we expand on the trRIXS definition: “As detailed in Methods, trRIXS is a photon-in-photon-out x-ray scattering process involving an intermediate state with finite lifetime. Due to the spin-orbit coupling at the core level (e.g., the $2p$ orbitals for the transition-metal L -edge RIXS), this intermediate state can involve spin flip events and couple to magnetic excitations of the valence band [77]. Therefore, trRIXS is sensitive to spin excitations and can be used to probe the nonequilibrium spin dynamics of light-driven materials [78].”

8. In the beginning of Sec.II B, we explained the style of trRIXS spectra suggested by the Referee: The trRIXS cross-section, denoted as $\mathcal{I}(q, \omega_i - \omega, \omega_i, t)$ in Methods, depends

on energy, momentum, and polarization of both the incident and scattered photons. In our simulation, we select a scattering geometry with π -polarized (parallel to the scattering plane) incident photons and σ -polarized (perpendicular to the scattering plane) scattered photons, which maximizes the spin-flip contribution to the trRIXS cross-section [77,78]. Due to the focus on spin entanglement, we keep this polarization configuration fixed throughout the paper and omit the polarization subscripts in \mathcal{I} . The trRIXS spectrum comes with two energy axes for the incident photon energy ω_i and the energy loss ω (difference between incident and scattering photon energies).” We further moved the important notions from the Methods to the main text, after Eq. (6): “Here, $\hat{D}_{q_i \epsilon_i}$ and $\hat{D}_{q_s \epsilon_s}^\dagger$ denote the dipole-transition operators for the incident and scattering process, respectively; $M^{(\text{in})}$ and $M^{(\text{out})}$ indicate the corresponding matrix elements and τ_{core} is the core-hole lifetime. The definition of these factors and the full expression of the trRIXS cross-section $\mathcal{I}(q, \omega_i - \omega, \omega_i, t)$ are explained in the Methods section.”

9. In the 1st paragraph of Sec. IIC (page 4) we motivate our choice of the extended-Hubbard model “Given prior equilibrium RIXS experiments [80], we consider 1D cuprate chains as an ideal platform for the experimental verification of our results. Recent experiments in $\text{Ba}_{2-x}\text{Sr}_x\text{CuO}_{3+\delta}$ have identified the EHM with mixed-sign interactions [72] as their underlying model Hamiltonian (see Methods) and here we investigate its light-driven dynamics. Throughout this paper, we set the on-site (U) and the nearest-neighbor (V) interactions to $U = 8t_h$ and $V = -t_h$, respectively, corresponding to the characteristic values for cuprate chains [72,81]. The existence of this nearest-neighbor term V is crucial for the presence of many-body entanglement, as we discuss in Sec. IID.”

10. In the 3rd paragraph of Sec.IIC (page 4) we emphasise our choice of q and lack of long-range order: “When calculating the time-dependent QFI, we focus on a specific momentum, $q = \pi/6$. Distinct from undoped antiferromagnets previously employed to witness equilibrium entanglement [56], this small-momentum wavevector exhibits the most evident spectral changes and captures the longest-range correlations supported by our system size. The 1D system simulated in this paper does not have a spontaneous symmetry breaking, which simplifies the entanglement analysis of the spin fluctuations.”

11. In Fig. 5d on page 5 we edited the label typo.

12. In the 1st paragraph on page 5, we clarify our theoretical analysis: “We then proceed to evaluate the QFI as it would be done in real trRIXS experiments, i.e., by only assessing the sequence of trRIXS snapshots without additional theoretical knowledge about the state of the driven system.” and address the high-order derivatives in the 2nd paragraph on page 5: “We evaluate these high-order derivatives using finite difference methods, starting from the time sequence of trRIXS spectra, and then solving the self-consistent equations.”

13. In the 1st paragraph on page 6, we discuss the non-negligible core-hole lifetime: “When the lifetime is not negligible, the conversion of the equilibrium RIXS intensity into the $S(q, \omega)$ requires a more systematic approach. One method consists of calculating the

four-particle response function as the lowest-order perturbative expansion of the lifetime τ_{core} [79]. For trRIXS, one can also correct for the finite lifetime effects by using an overall scaling factor determined by the equilibrium RIXS intensity and $S(q, \omega)$, both of which can be independently measured. As discussed in Sec. II of the SI, this correction provides a good approximation of the long-term dynamics even for a large core-hole lifetime.”

14. In the 1st paragraph of Sec.II D (page 6), we corrected Ref. 94 into “Lorenzana *et al.* Phys. Rev. B, 72, 224511 (2005)”

15. At the end of Sec. II.D (page 7), we emphasize the importance of the extended-Hubbard model: “In other words, the nearest-neighbor interaction, whose feature was recently identified in 1D cuprate chains but the phonon-mediated mechanism widely exists in transition-metal oxides, is crucial role towards achieving light control of entanglement in correlated materials.”

16. In the 2nd paragraph of the Discussion (page 7), we cite the recent spintronic arXiv paper. “Additionally, since the nonequilibrium control of entanglement was recently discussed also in the context of spintronic devices [100], our self-consistent trRIXS approach could eventually be applied to probe time-dependent entanglement in non-laser-driven systems.”

17. In the 1st paragraph on page 1 of our Supplementary Information we corrected the prefactor in Eq. (1) by adding “ $1/\sigma_{\text{pr}}^2$ ”.

18. In the 1st paragraph of the 1st page of the Supplementary Information (after Eq. (1)), we added an extension out of equilibrium to account for the thermal ensemble average: “Here, the average notation is not restricted to a pure state, but can also be generalized into a thermal ensemble [...Eq. (2)...] The $\hat{U}(t_1, t_2)$ is the unitary time evolution operator, \mathcal{Z} is the partition function of the equilibrium state (at $t = -\infty$), and β is the inverse temperature.”

19. In the 2nd paragraph on page 1 of our Supplementary Information, we explain the change of variable “By introducing a change of variable, $x = \tau - t$ we can Taylor expand $f_Q(\tau) = f_Q(t + x)$ around t in Eq. 4 to get...”

20. In the 2nd paragraph on page 1 of our Supplementary Information we fixed the expression for \mathcal{C}_m : “ $\mathcal{C}_m = -(1/\sigma_{\text{pr}}\sqrt{\pi}) \int_{-\infty}^{\infty} e^{-x^2/\sigma_{\text{pr}}^2} x^{2m} dx = -(\sigma_{\text{pr}}^{m-1/2}/\sqrt{\pi})\Gamma(m + 1/2)$.”

21. In the 2nd page of the Supplementary Information, we have added a new section “Correction of trRIXS with Long Core-Hole Lifetime”.

22. In the 3rd page of the Supplementary Information, we have added a new section “Sec. IV. Nonequilibrium Entanglement Entropy” to compare the entanglement entropy with the QFI dynamics.

23. We have additional minor changes, not directly corresponding to the Referees' requests. These changes include adjusting the aspect ratios for Fig. 3-5 and the text of all paragraphs for consistency and length consideration.

REVIEWER COMMENTS

Reviewer #1 (Remarks to the Author):

First of all, I want to thank the authors for their comprehensive response and improvements. My main concerns with the initial manuscript, namely the validity at finite temperature and/or long-range order, have been addressed in a convincing manner. (For the $T>0$ aspect the related work Phys. Rev. Lett. 130, 106902 (2023) was an important factor in convincing me.) Given that these issues have been resolved, I believe the manuscript is suitable for publication in Nature Communications after remaining comments have been addressed.

Below, I address the authors' responses to my questions, using the same numbering as in the rebuttal.

(1) *“First, we would like to stress that our work builds upon a general result for equilibrium systems by Hauke et al. (Nat. Phys. 12, 778), which rigorously connects the QFI for local operators with the equilibrium spectral response functions. Its extension to the nonequilibrium structure factor in Sec. II.A of the main text is as general as the Hauke’s conclusions, and only makes use of the spin operators as a concrete experimental example and a typical choice in quantum information.”*

I did not question the generality of the equilibrium work by Hauke et al., but did initially question whether the extension reported in this manuscript was as general. I do believe the authors have established this in the revised version. Nevertheless, the presentation could be clarified further. Some suggested improvements will follow under point (2).

I want to thank the authors for the addition of the entanglement entropy plot. While I appreciate the desire to focus the narrative on the QFI, it is a quantity few practicing condensed matter physicists have any intuition for. Hence, to build understanding of how to interpret it in practice, I do believe it is important that the QFI is not studied “in a vacuum”, especially when applied to new systems. Including a comparison with the entanglement entropy in supplemental material strikes a good balance in this regard.

(2) I do appreciate the large computational cost involved in simulating finite-temperature trRIXS spectra, and think it is reasonable to avoid doing so here. The related work [Phys. Rev. Lett. 130, 106902 (2023)] along with Fig. R2 do alleviate my concerns about using a $T=0$ simulation to approximate a $T>0$ experiment. However, I do still think the presentation is inconsistent with the stated goal of proposing an experimental method to measure out-of-equilibrium entanglement, which I do consider the potentially most impactful contribution of the paper. Instead, the presentation does at times seem tailored to only $T=0$ computations. I think it should be stated that Eq. (1) in the main text is the pure-state definition, but that it can be generalized to finite temperature such that the self-consistent equations are valid also at finite temperature. If the authors do not want to elaborate on the mixed-state case in the main text, it might be appropriate to refer readers to the supplemental material, to which several of the comments made in the response could be added.

“First, we stress that the QFI can be extracted from the trRIXS cross-section both in the case of pure and mixed initial states. While our numerical simulations are performed using a pure initial state, the self-consistent equations reported in Eqs. (4) and (5) are also valid for a thermal ensemble.”

Surely the manuscript would benefit significantly from including a statement like this?

(3) I consider this matter closed, but would recommend including a similar statement in the supplemental material.

(4) Please double check C_m again. The first equality is correct, but for the second I get $C_m = \frac{\sigma_{pr}^{2m}}{\sqrt{\pi}} \Gamma(m+1/2)$. The other corrections are fine.

(5) This is a useful clarification. However, I note that the beginning of the second sentence in paragraph 2 on page 1 has disappeared.

(6) *“While the connection between QFI and multiparticle entanglement is well established for spin operators in magnetic materials, analytical formulations for general fermionic systems are still lacking [70,71].”*

This is a clear improvement, and the statement is technically true. However, it may be worth noting that the construction of Hauke et al. is equally valid for fermionic lattice systems as it is for spin systems. The case of indistinguishable particles addressed in the two cited papers is indeed a different story.

(7)-(11) I consider these matters closed. I have verified that the equations in the SI are now consistent with the main text.

Additional comments:

Eq. (5) and SI Eq. (6): I think it is better to write $(2m)!$ than $2m!=2(m!)$.

SI Sec. II is a nice addition.

Reviewer #2 (Remarks to the Author):

The authors have answered all the questions satisfactorily and the manuscript can now be accepted for publication.

Reply to Reviewer #1

We thank the Reviewer for the positive evaluation of our work and are pleased that they state “*Given that these issues have been resolved, I believe the manuscript is suitable for publication in Nature Communications after remaining comments have been addressed.*” Again, we indicate in **blue** the original reports, in **black** our reply, and in **red** our changes in the revised manuscript as we address the remaining questions.

(1) Reviewer’s comment: “*(Quoted from the our first-round reply) First, we would like to stress that our work builds upon a general result for equilibrium systems by Hauke et al. (Nat. Phys. 12, 778), which rigorously connects the QFI for local operators with the equilibrium spectral response functions. Its extension to the nonequilibrium structure factor in Sec. II.A of the main text is as general as the Hauke’s conclusions, and only makes use of the spin operators as a concrete experimental example and a typical choice in quantum information.*” I did not question the generality of the equilibrium work by Hauke et al., but did initially question whether the extension reported in this manuscript was as general. I do believe the authors have established this in the revised version. Nevertheless, the presentation could be clarified further. Some suggested improvements will follow under point (2).

I want to thank the authors for the addition of the entanglement entropy plot. While I appreciate the desire to focus the narrative on the QFI, it is a quantity few practicing condensed matter physicists have any intuition for. Hence, to build understanding of how to interpret it in practice, I do believe it is important that the QFI is not studied “in a vacuum”, especially when applied to new systems. Including a comparison with the entanglement entropy in supplemental material strikes a good balance in this regard.”

Our reply: We thank the Reviewer’s positive evaluation of our revision. We agree that a comparison with entanglement entropy is helpful for most non-expert readers to understand the quantification of entanglement. We also appreciate the Reviewer’s first-round comments, which have helped us improve the narrative.

Changes made: We have improved the presentation in multiple aspects as specified in each of the following questions.

(2) Reviewer’s comment: “I do appreciate the large computational cost involved in simulating finite-temperature trRIXS spectra, and think it is reasonable to avoid doing so here. The related work [Phys. Rev. Lett. 130, 106902 (2023)] along with Fig. R2 do alleviate my concerns about using a $T = 0$ simulation to approximate a $T > 0$ experiment. However, I do still think the presentation is inconsistent with the stated goal of proposing an experimental method to measure out-of-equilibrium entanglement, which I do consider the potentially most impactful contribution of the paper. Instead, the presentation does at times seem tailored to only $T = 0$ computations. I think it should be stated that Eq. (1) in the main text is the pure-state definition, but that it can be generalized to finite temperature such that the self-consistent equations are valid also at finite temperature. If the authors do not want to elaborate on the mixed-state case in the main text, it might be appropriate to refer readers to the supplemental material, to which several of the comments made in the response could be added.

(Quoted from the our first-round reply) First, we stress that the QFI can be extracted

from the trRIXS cross-section both in the case of pure and mixed initial states. While our numerical simulations are performed using a pure initial state, the self-consistent equations reported in Eqs. (4) and (5) are also valid for a thermal ensemble.’ Surely the manuscript would benefit significantly from including a statement like this?”

Our reply: We agree with the Reviewer that the generality of our method in thermal ensembles should be emphasized before discussing our numerical simulations (Sec. IV). We also agree that the choice of a pure initial state should be rationalized in terms of computational convenience without implications for the generality of our theory.

Changes made: To clarify the generality of our approach, we have extended the definition of transient QFI as “The transient QFI $f_Q(q, t)$ is completely determined by the instantaneous wavefunction $|\psi(t)\rangle$, or an ensemble represented by the density matrix $\rho(t)$.” In the end of Sec. II, we further stressed the generality of our method and referred readers to the SI: “While we use the pure-state notation in the derivation of the QFI sum rule and choose a pure initial state in our simulations, this approach applies to both pure and mixed initial states (see Sec. I of SI for further details). This generalization relies on considering $\langle \dots \rangle$ as an ensemble average and on the linearity of Eqs. (4) and (5).” Moreover, we have restated the choice of a pure state before presenting the numerical results in the second paragraph of Sec. II C: “We employ the ground state at zero temperature as the initial state, due to the computational complexity of simulating the trRIXS cross-section of an ensemble. The generality of this method is further discussed in Sec. II A and the SI.”

Finally, we have expanded the ensemble generalization discussion in Sec. I of the Supplemental Information: “In this paper, we employ a pure initial state $|\psi(-\infty)\rangle$ at zero temperature for all simulations, with the aim to reduce the computational complexity and focus on the nonequilibrium aspect of the problem. However, the expressions for the QFI and the trRIXS spectra, as well as the relation between them [i.e. Eq. (6) of the main text] are not restricted to a pure state, but can be generalized to any mixed initial state with time-independent distribution weights. Guaranteed by the linearity of the Eq. (S1), the average notation can be generalized to represent a thermal ensemble”.

(3) Reviewer’s comment: “I consider this matter closed, but would recommend including a similar statement in the supplemental material.”

Our reply: We thank the Reviewer for the suggestion. We have emphasized the discussion about the disconnected QFI part and elastic peaks by a similar statement in the Sec. I of our Supplemental Information.

Changes made: To clarify the role of the disconnected part, we have added a brief statement in the Sec. I of SI: “... the QFI $f_Q(q, t)$ is defined as $4\langle \hat{\rho}_{-q}^s(t)\hat{\rho}_q^s(t) \rangle/N$, when the SU(2) symmetry is preserved. (The disconnected part of Eq. (1) in the main text is non-zero in the presence of long-range magnetic order; however, it can be evaluated separately from the elastic scattering peak intensity and subtracted off from the correlation function). The expression for $f_Q(q, t)$ is an equal-time measurement ...”.

(4) Reviewer’s comment: “Please double check C_m again. The first equality is correct, but for the second I get $C_m = -\frac{\sigma_{pr}^{2m}}{\sqrt{\pi}}\Gamma(m + 1/2)$. The other corrections are fine.”

Our reply: We thank the Reviewer for bringing the C_m typo to our attention. We have

corrected the two instances of this error in the main text and the Supplemental Information.

Changes made: We have edited the corrected the expression of C_m just below main text Eq. (5) and SI Eq. (S6) to be $C_m = -\frac{\sigma_{pr}^{2m}}{\sqrt{\pi}}\Gamma(m + \frac{1}{2})$

(5) Reviewer’s comment: “This is a useful clarification. However, I note that the beginning of the second sentence in paragraph 2 on page 1 has disappeared.”

Our reply: We thank the Reviewer for pointing of this compiling issue. This issue did not appear in our original version with line numbers. However, the L^AT_EX macro used for the first revised sentence was broken when we turned off the line-number command.

Changes made: We have removed the macro and retyped the second sentence of the second paragraph.

(6) Reviewer’s comment: “(Quoted from the manuscript) *While the connection between QFI and multiparticle entanglement is well established for spin operators in magnetic materials, analytical formulations for general fermionic systems are still lacking [70,71].*” This is a clear improvement, and the statement is technically true. However, it may be worth noting that the construction of Hauke et al. is equally valid for fermionic lattice systems as it is for spin systems. The case of indistinguishable particles addressed in the two cited papers is indeed a different story.”

Our reply: We thank the reviewer for pointing out the construction of Hauke *et al.* is equally valid for fermion lattice systems as elaborated in Ref. 4 by de Almeida and Hauke that we have followed in the Supplementary Information. Ref. 4 defines the basis-dependent mode multipartite entanglement in fermionic lattice systems by replacing the summation of spin operators with a weighted summation of fermion operators, as an important extension of the QFI usage. With indistinguishable particles, the particle entanglement acquires a different meaning. Entanglement becomes an intrinsic, basis-independent property of the many-body wavefunction which can be characterized by more complicated quantities like the Slater rank. (Thus, a non-interacting Fermi sea, which is separable in the momentum representation, is still non-entangled in the real-space representation). In the context of the introduction, we meant to refer to doped fermionic systems, which have not yet been studied and are not relevant for indistinguishable particles; in the context of the end of Sec. II D, we meant to refer to the difficulty of mapping the QFI (and spectrum) to the basis-independent entanglement witness for indistinguishable fermions. To avoid confusion, we have revised these two sections of the main text.

Changes made: In the end of the introduction section, we have revised the discussion of the 3rd challenge: “**While the connection between QFI and multiparticle entanglement is well established for spin operators in magnetic materials and gapped fermionic lattice modes [70], this connection is unclear for interacting fermions with dopant carriers, which is particularly relevant to the study of light-driven materials.**” In addition, we further added the following remark at the end of Sec. II D, “**A comprehensive study of basis-independent entanglement witnesses for indistinguishable fermions is beyond the scope of this work.**” We have also added Ref. [70] (de Almeida *et al.*) in the same sentence of the main text, to acknowledge that our fermionic mode discussions in the SI follows this study’s approach.

(7)-(11) Reviewer’s comment: I consider these matters closed. I have verified that the

equations in the SI are now consistent with the main text.

Our reply and changes: No changes.

(12) Reviewer’s comment: “Eq. (5) and SI Eq. (6): I think it is better to write $(2m)!$ than $2m!=2(m!).$ ”

Our reply: We agree with the Reviewer that this change would make the notation clearer in Eq. (5) and SI Eq. (S6).

Changes made: We have changed the notation from $2m!$ to $(2m)!$ to avoid ambiguity.

Summary of Changes

All revisions as recommended by the Reviewers have been marked **red** in the resubmitted manuscript. Here we summarize the changes:

1. In the 2nd paragraph on the 1st page, we complete the missing half sentence: “**Entangled wavefunctions in synthetic few-body quantum simulators can ...**”

2. In the 1st paragraph on the 2nd page, we discuss the: “**While the connection between QFI and multiparticle entanglement is well established for spin operators in magnetic materials and gapped fermionic lattice modes [70], this connection is unclear for interacting fermions with dopant carriers, which is particularly relevant to the study of light-driven materials.**”

3. In the 1st paragraph of Sec. II A (page 2), we extend the definition of transient QFI: “**The transient QFI $f_Q(q, t)$ is completely determined by the instantaneous wavefunction $|\psi(t)\rangle$, or an ensemble represented by the density matrix $\rho(t)$.**”

4. In Eq. (5) on page 3 [and SI Eq. (S8)], we have changed the notation from $2m!$ to $(2m)!$.

5. In the 2nd paragraph on page 3 (below Eq. (5) [and SI Eq. (S6)]), we fixed the expression for C_m : “ **$C_m = -\frac{\sigma_{pr}^{2m}}{\sqrt{\pi}} \Gamma\left(m + \frac{1}{2}\right)$.**”

6. In the last paragraph of Sec. II A (page 3), we explain the generality of our method: “**While we use the pure-state notation in the derivation of the QFI sum rule and choose a pure initial state in our simulations, this approach applies to both pure and mixed initial states (see Sec. I of SI for further details). This generalization relies on considering $\langle \dots \rangle$ as an ensemble average and on the linearity of Eqs. (4) and (5).**”

7. In the 2nd paragraph of Sec. II C (page 4), we explain our choice of a pure state: “**We employ the ground state at zero temperature as the initial state, due to the computational complexity of simulating the trRIXS cross-section of an ensemble. The generality of this method is further discussed in Sec. II A and the SI.**”

8. In the last paragraph of Sec. II D (page 7), we promote the reference in SI into the main text as “we report an extension of our calculations to the single-particle fermionic modes of the driven EHM following the approach in Ref. 70, ...” We also clarify the distinction from indistinguishable fermions as “A comprehensive study of basis-independent entanglement witnesses for indistinguishable fermions is beyond the scope of this work.”

9. In the 1st paragraph on the 1st page of the Supplementary Information we expand the ensemble generalization discussion: “In this paper, we employ a pure initial state $|\psi(-\infty)\rangle$ at zero temperature for all simulations, with the aim to reduce the computational complexity and focus on the nonequilibrium aspect of the problem. However, the expressions for the QFI and the trRIXS spectra, as well as the relation between them [i.e. Eq. (6) of the main text] are not restricted to a pure state, but can be generalized to any mixed initial state with time-independent distribution weights. Guaranteed by the linearity of the Eq. (S1), the average notation can be generalized to represent a thermal ensemble”

10. In the 1st paragraph on the 1st page of the Supplementary Information we clarify the role of the disconnected part: “... the QFI $f_Q(q, t)$ is defined as $4\langle\hat{\rho}_{-q}^s(t)\hat{\rho}_q^s(t)\rangle/N$, when the SU(2) symmetry is preserved. (The disconnected part of Eq. (1) in the main text is non-zero in the presence of long-range magnetic order; however, it can be evaluated separately from the elastic scattering peak intensity and subtracted off from the correlation function). The expression for $f_Q(q, t)$ is an equal-time measurement ...”